# JMJD6 is a tumorigenic factor and therapeutic target in neuroblastoma

Matthew Wong [1,18], Yuting Sun[1,18], Zhichao Xi [2,3,4,18], Giorgio Milazzo [5], Rebecca C. Poulos [6,7], Christoph Bartenhagen[8], Jessica L. Bell [9], Chelsea Mayoh [1], Nicholas Ho [1], Andrew E. Tee[1], Xiaoqiong Chen[2,4], Yang Li[2,4], Roberto Ciaccio [5], Pei Y. Liu [1], Chen C. Jiang [10], Qing Lan[11], Nisitha Jayatilleke [1], Belamy B. Cheung[1], Michelle Haber [1], Murray D. Norris [1,12], Xu D. Zhang [10], Glenn M. Marshall[1,13], Jenny Y. Wang[1], Stefan Hüttelmaier [9], Matthias Fischer[8], Jason W.H. Wong [6,14], Hongxi Xu [2,4], Giovanni Perini[5], Qihan Dong[3,15], Rani E. George[16,17] & Tao Liu [1]

Chromosome 17q21-ter is commonly gained in neuroblastoma, but it is unclear which gene in the region is important for tumorigenesis. The *JMJD6* gene at 17q21-ter activates gene transcription. Here we show that JMJD6 forms protein complexes with N-Myc and BRD4, and is important for E2F2, N-Myc and c-Myc transcription. Knocking down JMJD6 reduces neuroblastoma cell proliferation and survival in vitro and tumor progression in mice, and high levels of JMJD6 expression in human neuroblastoma tissues independently predict poor patient prognosis. In addition, *JMJD6* gene is associated with transcriptional super-enhancers. Combination therapy with the CDK7/super-enhancer inhibitor THZ1 and the histone dea-cetylase inhibitor panobinostat synergistically reduces JMJD6, E2F2, N-Myc, c-Myc expression, induces apoptosis in vitro and leads to neuroblastoma tumor regression in mice, which are significantly reversed by forced JMJD6 over-expression. Our findings therefore identify JMJD6 as a neuroblastoma tumorigenesis factor, and the combination therapy as a treatment strategy.

[1] Children's Cancer Institute Australia, Randwick Sydney, NSW 2031, Australia. [2] School of Pharmacy, Shanghai University of Traditional Chinese Medicine, Shanghai 201203, China. [3] Central Clinical School and Bosch Institute, The University of Sydney, Sydney, NSW 2006, Australia. [4] Institute of Cardiovascular Disease of Integrated Traditional Chinese and Western Medicine, Shuguang Hospital, Shanghai University of Traditional Chinese Medicine, Shanghai 201203, China. [5] Department of Pharmacy and Biotechnology, University of Bologna, 40126 Bologna, Italy. [6] Prince of Wales Clinical School and Lowy Cancer Research Centre, UNSW Australia, Sydney, NSW 2052, Australia. [7] Children's Medical Research Institute Faculty of Medicine and Health, The University of Sydney, Westmead, NSW 2145, Australia. [8] Department of Experimental Pediatric Oncology, University Hospital, University of Cologne, 50931 Cologne, Germany. [9] Institute of Molecular Medicine, Martin Luther University, Kurt-Mothes-Str.3a, 06120 Halle Saale, Germany. [10] School of Biomedical Sciences and Pharmacy, The University of Newcastle, Newcastle, NSW 2308, Australia. [11] Department of Neurosurgery, the Second Affiliated Hospital of Soochow University, Suzhou 215004 Jiangsu, China. [12] Centre for Childhood Cancer Research, UNSW Medicine, UNSW Sydney, Kensington, Sydney, NSW 2052, Australia. [13] Kids Cancer Centre, Sydney Children's Hospital, Randwick, NSW 2031, Australia. [14] School of Biomedical Sciences, Li Ka Shing Faculty of Medicine, The University of Hong Kong, Hong Kong Special Administrative Region 999077, China. [15] School of Science and Health, The University of Western Sydney, Sydney, NSW 2751, Australia. [16] Department of Pediatric Hematology and Oncology, Dana-Farber Cancer Institute and Boston Children's Hospital, Boston, MA 02215, USA. [17] Department of Pediatrics, Harvard Medical School, Boston, MA 02115, USA. [18] These authors contributed equally: Matthew Wong, Yuting Sun, Zhichao Xi. Correspondence and requests for materials should be addressed to H.X. (email: xuhongxi88@gmail.com) or to T.L. (email: tliu@ccia.unsw.edu.au)

Neuroblastoma is the most prevalent solid tumor in early childhood, and accounts for ~15% of all childhood cancer death[1]. Unlike adult cancers, the most prominent features of human neuroblastoma are gene deletion, amplification, or gain[1].

The most common gene copy number variation in human neuroblastoma is chromosome 17q21-ter gain, which predicts poor patient prognosis[2]. The genes *survivin* and *insulin growth factor 2 binding protein 1* within this chromosomal region have been shown to promote neuroblastoma cell survival and increase *MYCN* mRNA expression, respectively[3,4]. However, it is not clear which genes at 17q21-ter are critical for neuroblastoma tumorigenesis.

The jumonji domain-containing 6 (JMJD6) gene at chromosome 17qter is a dual arginine demethylase and lysyl hydroxylase of histone and nonhistone proteins[5,6]. As a histone arginine demethylase, JMJD6 upregulates target gene transcription by forming a protein complex with BRD4 and demethylating histone H4 at arginine 3 (H4R3) at target gene antipause enhancers, leading to RNA polymerase II (RNA Pol II) release from promoter-proximal pause regions and aberrant gene expression in glioblastoma[7,8]. As a lysyl hydroxylase, JMJD6 forms a protein complex with p53 and catalyzes p53 protein hydroxylation, leading to p53 inactivation, colon cancer cell proliferation and survival[9]. As such, targeting the demethylase or hydroxylase activity of JMJD6 alone is not an ideal strategy for effective cancer therapy.

One of the most promising anticancer strategies is to suppress oncogene transcription by disrupting associated transcriptional super-enhancers. Transcriptional super-enhancers are found selectively at the loci of oncogenes and are therefore ideal targets for cancer therapy[10,11]. THZ1, a specific covalent inhibitor of CDK7, selectively blocks the transcription of super-enhancer-associated oncogenes, such as *MYC, MYCN, YAP1*, and *RUNX1*, thereby inducing cancer cell growth inhibition and apoptosis. In this regard, THZ1 has been shown to inhibit leukemia, lung cancer, *MYCN* gene-amplified neuroblastoma, and esophageal squamous cell carcinoma progression in mice[12–15].

In this study, we have found that the *JMJD6* gene is gained in ~80% of human neuroblastoma tissues. JMJD6 formed protein complexes with N-Myc and the transcriptional coactivator BRD4. Suppression of JMJD6 resulted in decreased E2F2, N-Myc, and c-Myc gene expression, reduced neuroblastoma cell proliferation, induction of apoptosis in vitro, and inhibition of tumor growth in vivo. While JMJD6 is associated with super-enhancers, treatment with THZ1 and the histone deacetylase (HDAC) inhibitor panobinostat synergistically reduced JMJD6, E2F2, N-Myc, and c-Myc expression, induced tumor cell apoptosis in vitro, and led to neuroblastoma tumor regression in mice, which were significantly reversed by forced JMJD6 overexpression, suggesting this combination as a therapeutic approach for neuroblastoma.

## Results

### High *JMJD6* in neuroblastomas predicts poor patient prognosis.
The *JMJD6* gene, located at chromosome 17qter, is one of the few histone modification genes located within the 17q21-ter region. We analyzed the human neuroblastoma array comparative genomic hybridization (array-CGH) datasets from 209 patients[16–18] and the matched tumor tissue microarray gene expression Kocak dataset including the 209 patients[18], both published previously and downloaded from the Gene Expression Omnibus website of the National Center for Biotechnology Information (SuperSeries GSE45480). Analysis of the datasets showed that the *JMJD6* gene was gained in every human neuroblastoma tissue with 17q gain, and that chromosome 17q/*JMJD6* gene was gained in 172 of the

total of 209 (82.30%), including 21 of 26 *MYCN* amplified (80.77%) and 149 of 181 *MYCN*-non-amplified (82.32%), human neuroblastoma tissues (Fig. 1a, b). For the two human neuroblastoma tissues with unknown *MYCN* amplification status, 17q/*JMJD6* gene was gained in both of the samples.

We next examined whether chromosome 17q segmental gain and numerical (whole chromosome) gain had different effects on 17q copy number and *JMJD6* gene expression. While tumors with 17q segmental gain or numerical gain showed higher 17q copy number than tumors without 17q gain, tumors with 17q segmental gain also showed significantly higher 17q copy number than tumors with 17q numerical gain (Fig. 1c). Consistent with these data, *JMJD6* gene expression was significantly higher in tumors with 17q segmental gain than in tumors with 17q numerical gain or without 17q gain, and there was no difference in *JMJD6* gene expression between tumors with 17q numerical gain and tumors without 17q gain (Fig. 1d).

To assess the clinical relevance of JMJD6 in human neuroblastoma tissues, expression of JMJD6, as well as N-Myc and c-Myc, was examined in the publicly available microarray gene expression Versteeg[19] and Oberthuer[18,20] datasets consisting of 88 and 476 human neuroblastoma patient samples with prognosis information, respectively. Two-sided Pearson's correlation study showed that JMJD6 mRNA expression positively correlated with N-Myc mRNA expression in the 88 and 476 human neuroblastoma tissues of the Versteeg and Oberthuer datasets (Supplementary Fig. 1a), and that JMJD6 mRNA expression positively correlated with c-Myc mRNA expression in the 405 *MYCN*-non-amplified human neuroblastoma tissues of the large Oberthuer dataset (Supplementary Fig. 1b). Since N-Myc and c-Myc mutually suppress each other's expression, and N-Myc and c-Myc mRNA expression is well known to inversely correlate with each other in human neuroblastoma tissues[21], we used the higher value of N-Myc and c-Myc expression as the N-Myc/c-Myc expression value (two Myc as a group) for each of the 476 human neuroblastoma tissues. Two-sided Pearson's correlation study revealed that JMJD6 mRNA expression positively correlated with N-Myc/c-Myc mRNA expression with a good effect size ($R = 0.447$) in the 476 neuroblastoma tissues (Fig. 1d).

Kaplan–Meier survival analysis showed that high levels of JMJD6 mRNA expression in human neuroblastoma tissues were associated with poor prognosis in the 88 and the 476 neuroblastoma patients of the Versteeg and the Oberthuer datasets, respectively ($p < 0.0001$) (Fig. 1e). Furthermore, high levels of JMJD6 mRNA expression in the 405 *MYCN*-non-amplified and the 66 *MYCN*-amplified neuroblastoma tissues were also positively associated with poor patient overall survival in the large Oberthuer dataset (Fig. 1f). Importantly, using the median or upper quartile of JMJD6 mRNA expression as the cut-off points, multivariable Cox regression analysis showed that high levels of JMJD6 mRNA expression was associated with poor patient overall survival and event-free survival, independent of disease stage, age at the time of diagnosis and *MYCN* amplification status (Table 1), and the current key prognostic markers for neuroblastoma patients[22].

In comparison, Kaplan–Meier survival analysis of the 209 human neuroblastoma patients with array-CGH data showed that chromosome 17q gain, when samples with segmental or numerical gain were combined, did not associate with patient overall survival (Supplementary Fig. 1c). When tumors with chromosome 17q segmental gain or numerical gain were analyzed separately, 17q segmental gain was associated with poor patient overall survival, whereas 17q numerical gain was not associated with patient overall survival (Supplementary Fig. 1d). In addition, the *MXRA7* gene is immediately upstream of the *JMJD6* gene at chromosome 17qter and gained in the same manner as the *JMJD6*

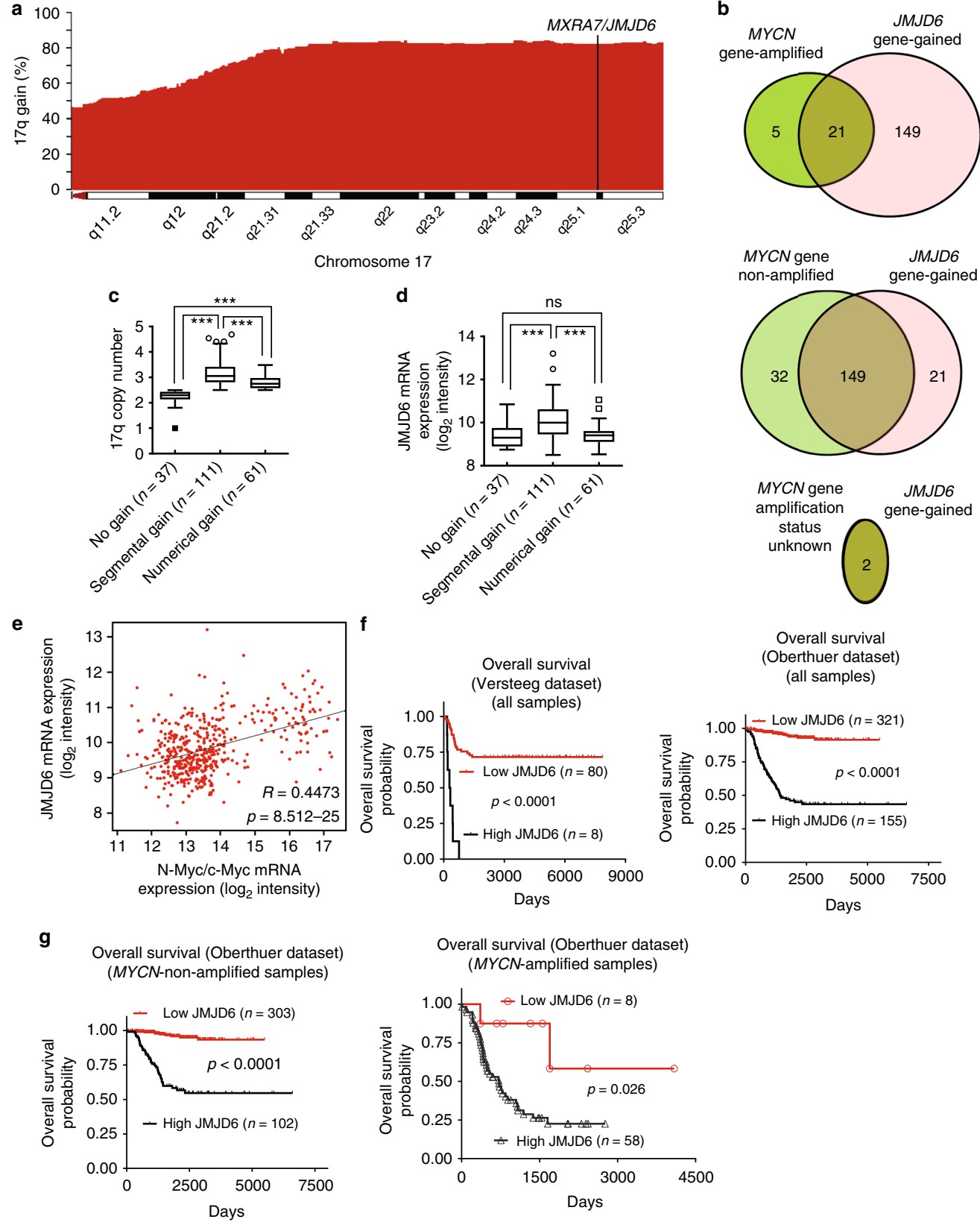

gene (Fig. 1a). Using the optimal Kaplan scanning, the median, upper or lower quartile of MXRA7 mRNA expression as the cut-off points, Kaplan–Meier survival analysis showed that high levels of MXRA7 mRNA expression in human neuroblastoma tissues were not associated with poor overall survival, but were

associated with better prognosis, in the 88 and the 476 neuroblastoma patients of the Versteeg and the Oberthuer datasets, respectively (Supplementary Fig. 2a, b).

Taken together, the data suggest that the *JMJD6* gene is commonly gained in human neuroblastoma tissues, and that a

**Fig. 1** High *JMJD6* gene expression in neuroblastoma tissues predicts poor patient prognosis. **a, b** Array-CGH data were employed to examine chromosome 17q and *JMJD6* gene gain in 209 human neuroblastoma tissues. The frequencies of chromosome gain at different regions of chromosome 17q were plotted against genomic locations in the 209 neuroblastoma tissues **a**. The Venn diagram showed overlap between *JMJD6* gene gain and *MYCN* gene amplification or nonamplification in the human neuroblastoma tissues **b**. **c, d** Chromosome 17q copy number (**c**) and *JMJD6* gene expression (**d**) were examined among the 209 human neuroblastoma tissues according to 17q segmental gain, numerical gain, or no gain. *JMJD6* gene expression was obtained from the Kocak microarray gene expression dataset. The center line was the median, the ends of the box were the upper and lower quartiles, and the whiskers were set to the minimum and maximum value within ±1.5 × the interquartile range (***$p < 0.001$, ns: not significant, one-way ANOVA). **e** Two-sided Pearson's correlation was employed to analyze the correlation between JMJD6 and N-Myc/c-Myc mRNA expression in the 476 human neuroblastoma tissues of the microarray gene expression Oberthuer dataset. The higher value of N-Myc and c-Myc expression was used as the N-Myc/c-Myc expression value for each of the neuroblastoma tissues. **f, g** Kaplan–Meier curves showed the probability of overall survival of patients according to JMJD6 mRNA expression levels in the 88 and 476 neuroblastoma samples in the Versteeg and Oberthuer datasets **f**, or in the 66 *MYCN*-amplified and 405 *MYCN*-non-amplified neuroblastoma samples in the large Oberthuer dataset **g**, using the optimal cut-off level determined by Kaplan scanning and two-sided log-rank tests. Source data are provided as a Source Data file

| Table 1 Multivariable Cox regression analysis of factors prognostic for patient survival[a] | | | | |
|---|---|---|---|---|
| **Factors** | **Overall survival** | | **Event-free survival** | |
| | **HR (95% CI)** | ***p*-value** | **HR (95% CI)** | ***p*-value** |
| High JMJD6 expression (median level as the cut off) | 1.87 (1.16–3.02) | 0.010 | 1.52 (1.04–2.20) | 0.029 |
| MYCN amplification | 3.24 (2.05–5.11) | 6.1E-5 | 1.76 (1.18–2.62) | 0.005 |
| Age > 18 months | 3.44 (1.88–6.30) | 2.6E-05 | 1.75 (1.19–2.59) | 0.005 |
| Stages 3 and 4[b] | 3.14 (1.55–6.35) | 0.002 | 2.38 (1.57–3.62) | 4.8E-05 |
| High JMJD6 expression (upper quartile as the cut off) | 4.58 (2.15–9.74) | 7.8E-5 | 1.56 (1.06–2.29) | 0.024 |
| MYCN amplification | 3.19 (2.05–4.96) | 2.6E-7 | 1.82 (1.23–2.68) | 0.003 |
| Age > 18 months | 3.20 (1.77–5.78) | 1.2E-04 | 1.77 (1.21–2.60) | 0.004 |
| Stages 3 and 4[b] | 2.83 (1.41–5.66) | 0.003 | 2.38 (1.57–3.61) | 4.4E-05 |

[a]The level of JMJD6 expression was considered high or low in relation to the median or upper quartile level of expression in tumors of the Oberthuer dataset. Hazard ratios were calculated as the antilogs of the regression coefficients in the proportional hazards regression. Multivariable Cox regression analysis was carried out following the inclusion of the four above listed factors into the Cox regression model, and *p*-value was obtained from two-sided log-rank test
[b]Tumor stage was categorized as favorable (International Neuroblastoma Staging System stages 1, 2, and 4 S) or unfavorable (International Neuroblastoma Staging System stages 3 and 4). Source data are provided as a Source Data file

high level of *JMJD6* expression independently predicts poor patient prognosis.

**Myc upregulates JMJD6 by binding to its gene promoter**. Myc oncoproteins induce gene transcription by direct binding to canonical and noncanonical E-boxes at target gene promoters[23]. Our bioinformatics analysis identified canonical (CACGTG) (−186 to −181) and noncanonical (CACGCG) (−81 to −76bp) E-boxes upstream of the *JMJD6* gene transcription start site. We therefore examined whether N-Myc and c-Myc modulated *JMJD6* gene expression in CHP134 (chromosome 17q21-ter/*JMJD6* gained and *MYCN* amplified) and SK-N-AS (chromosome 17q21-ter/*JMJD6* gained and *MYCN*-non-amplified) neuroblastoma cells. As shown in Fig. 2a, b, transfection of CHP134 cells with N-Myc siRNA-1 or N-Myc siRNA-2 efficiently knocked down N-Myc mRNA and protein expression, and reduced JMJD6 mRNA and protein expression. Similarly, transfection of SK-N-AS cells with c-Myc siRNA-1 or c-Myc siRNA-2 efficiently knocked down c-Myc mRNA and protein expression, and reduced JMJD6 mRNA and protein expression (Fig. 2a, b and Supplementary Fig. 3a). Consistent with these findings, withdrawal of doxycycline (DOX) from cell culture media of SHEP Tet/21N cells, which are stably transfected with a tetracycline/DOX withdrawal-inducible N-Myc expression construct, increased N-Myc and JMJD6 mRNA and protein expression (Fig. 2c).

We next performed chromatin immunoprecipitation (ChIP) assays with an anti-N-Myc or anti-c-Myc antibody or control IgG, and real-time PCR with primers targeting the *JMJD6* gene region (Fig. 2d). The ChIP assays showed that the N-Myc and c-Myc antibodies efficiently immunoprecipitated the E-box

region, compared with the negative control or exon 2 region (Fig. 2e). Taken together, the data suggest that N-Myc and c-Myc upregulate *JMJD6* gene expression via binding to the *JMJD6* gene promoter.

**JMJD6 induces E2F2, Myc, and their target gene expression**. As an arginine demethylase, JMJD6 is known to induce target gene transcription by interacting with the transcriptional coactivator bromodomain protein BRD4 and demethylating histone H4R3 at target gene enhancers[7]. We performed protein co-immunoprecipitation assays with an anti-BRD4 or JMJD6 antibody or control IgG in CHP134 cells. Immunoblot analysis showed that the anti-BRD4 and anti-JMJD6 antibodies efficiently immunoprecipitated both BRD4 and JMJD6 proteins (Fig. 3a), suggesting that JMJD6 forms a protein complex with BRD4 in neuroblastoma cells.

We next established CHP134 and SK-N-AS cells stably expressing DOX-inducible control shRNA, JMJD6 shRNA-1 or JMJD6 shRNA-2 FH1tUTG construct (Fig. 3b)[24]. Affymetrix microarray experiments were performed in DOX-inducible control shRNA, JMJD6 shRNA-1, and JMJD6 shRNA-2 CHP134 cells 40 h after treatment with vehicle control or DOX. Differential expression analysis identified a number of genes upregulated or downregulated after *JMJD6* gene knockdown including E2F2, which was downregulated, following the induction of JMJD6 shRNA-1 and JMJD6 shRNA-2 (Supplementary Data 1). We also performed Affymetrix microarray experiments in DOX-inducible control shRNA and JMJD6 shRNA-2 SK-N-AS cells 40 h after treatment with vehicle control or DOX. Differential gene expression and gene set enrichment analysis

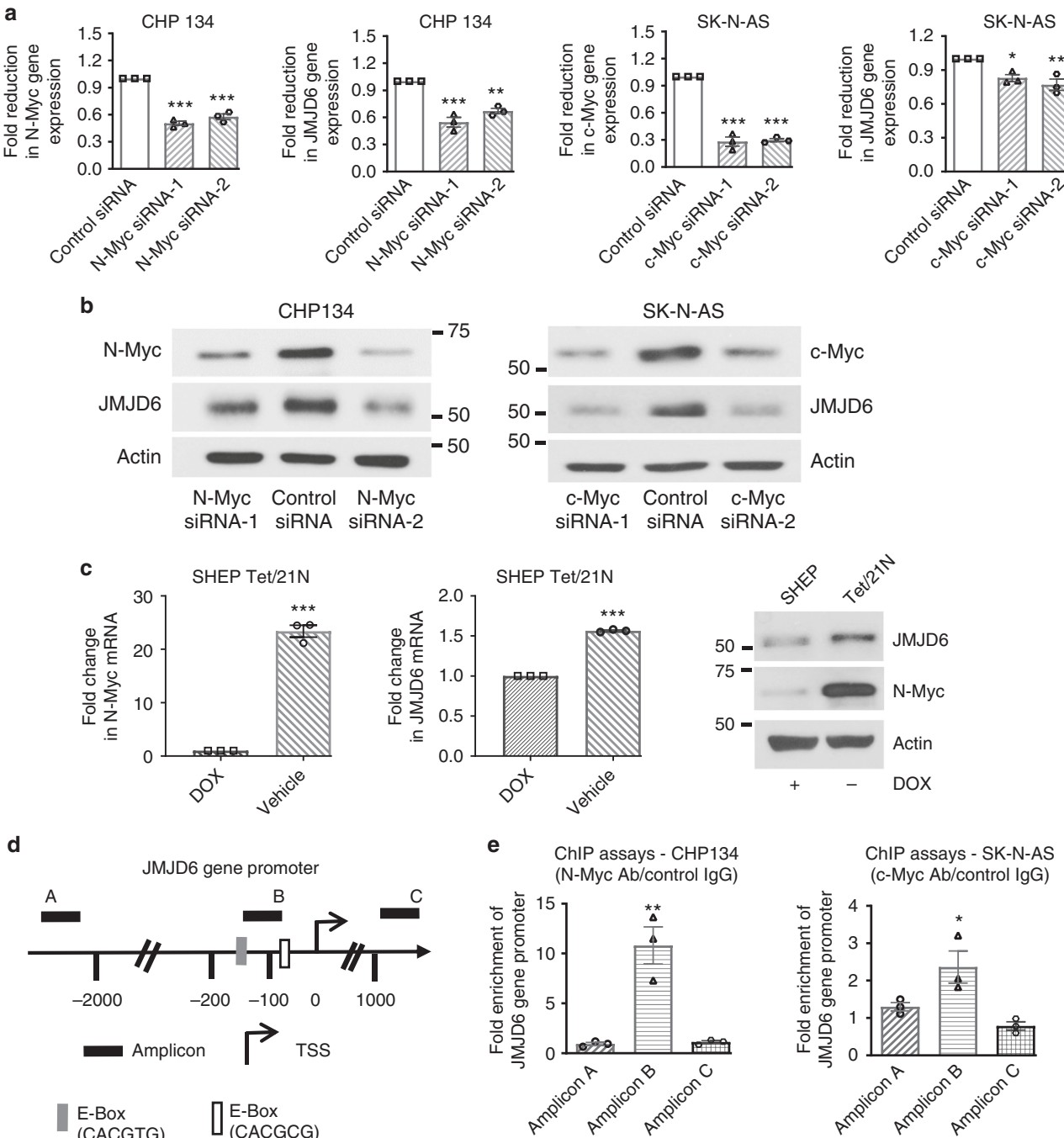

**Fig. 2** N-Myc and c-Myc upregulate JMJD6 expression by binding to the *JMJD6* promoter. **a**, **b** CHP134 cells were transfected with control siRNA, N-Myc siRNA-1 or N-Myc siRNA-2, and SK-N-AS cells were transfected with control siRNA, c-Myc siRNA-1, or c-Myc siRNA-2. Forty-eight hours later, RNA and protein were extracted for RT-PCR (**a**) and immunoblot (**b**) analyses. Error bars represent normalized standard errors from three independent experiments (**$p < 0.01$, ***$p < 0.001$, one-way ANOVA). **c** SHEP Tet/21N cells were treated with DOX (2 µg/ml) or vehicle control for 48 h. RT-PCR and immunoblot analyses were conducted. Error bars represent normalized standard errors from three independent experiments (***$p < 0.001$, two-tailed unpaired Student's $t$ test). **d** Schematic representation of the *JMJD6* gene promoter. TSS indicates transcription start site. Amplicons A, B, and C represented the sites for ChIP PCR primers. **e** ChIP assays were performed with a control IgG, anti-N-Myc or anti-c-Myc antibody (Ab), followed by PCR with primers targeting a remote negative control region (amplicon A), the E-box region (amplicon B), or the exon 2 region (amplicon C) of the *JMJD6* gene in CHP134 and SK-N-AS cells. Fold enrichment of the *JMJD6* gene region was calculated as the difference in cycle thresholds obtained with the anti-N-Myc or anti-c-Myc Ab compared with the control IgG, relative to input. Error bars represent normalized standard errors from three independent experiments (*$p < 0.05$, **$p < 0.01$, two-tailed unpaired Student's $t$ test). Source data are provided as a Source Data file

(GSEA) revealed that the transcription factor binding site consistently repressed after JMJD6 knockdown in both CHP134 and SK-N-AS cell lines was the binding site for E2F (Supplementary Tables 1, 2).

To examine whether JMJD6 directly regulates gene transcription, we performed ChIP sequencing (ChIP-Seq) experiments in DOX-inducible JMJD6 shRNA-2 CHP134 cells. The ChIP-Seq data identified a list of genes with considerably

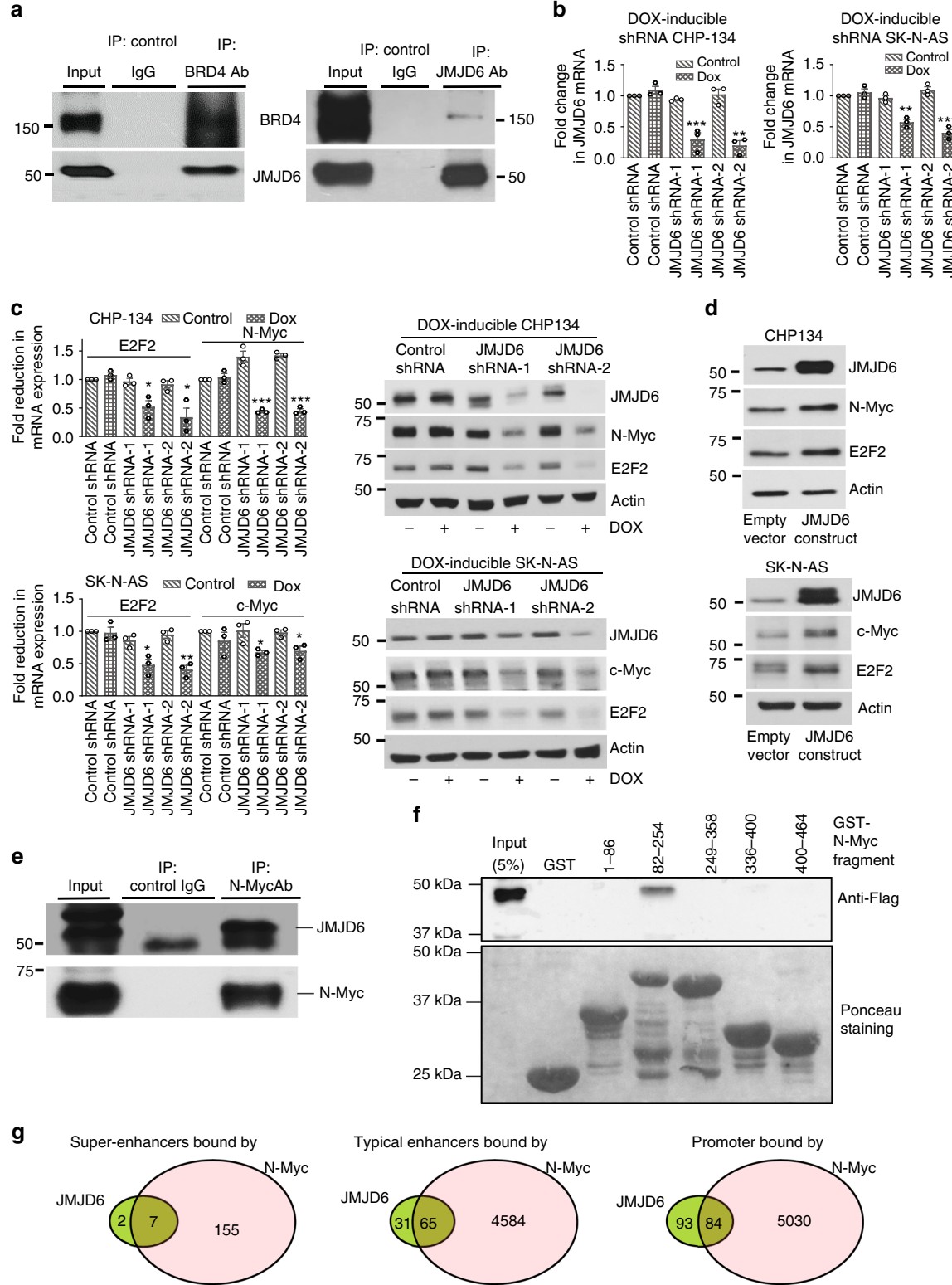

reduced RNA Pol II binding peaks at their gene promoters (Supplementary Data 2). GSEA analysis of the repressed genes showed that the top four gene sets were G2M checkpoint genes, E2F target genes, mitotic spindle genes, and Myc target genes (Supplementary Table 3). While the *MYCN* gene was not short-listed as one of the genes with loss of RNA Pol II binding peaks at promoters in Supplementary Data 2, ChIP-Seq and ChIP PCR data revealed 60% reduction in RNA Pol II binding at the

*MYCN* gene promoter after JMJD6 knockdown (Supplementary Fig. 3b–d).

We next examined whether JMJD6 regulated E2F2, N-Myc, and c-Myc expression. DOX-inducible control shRNA, JMJD6 shRNA-1 or JMJD6 shRNA-2 CHP134, and SK-N-AS cells were treated with vehicle control or DOX. Real-time reverse transcription PCR (RT-PCR) and immunoblot analyses confirmed that knocking down JMJD6 significantly reduced E2F2,

**Fig. 3** JMJD6 forms complexes with N-Myc and BRD4 to induce E2F2 and Myc expression. **a** Protein extracted from CHP134 cells was immunoprecipitated (IP) overnight with 2.5 μg of control IgG, anti-JMJD6 or anti-BRD4 antibody (Ab). The immunoprecipitated protein was immunoblotted with anti-JMJD6 or anti-BRD4 Ab. **b**, **c** DOX-inducible control shRNA, JMJD6 shRNA-1 or JMJD6 shRNA-2 CHP134, and SK-N-AS cells were treated with vehicle control or DOX for 48 h, followed by RT-PCR analysis of JMJD6 mRNA (**b**), or RT-PCR and immunoblot analysis of JMJD6, N-Myc, c-Myc, and E2F2 mRNA and protein (**c**). Error bars represent normalized standard errors from three independent experiments (*$p < 0.05$, **$p < 0.01$, ***$p < 0.001$, two-tailed unpaired Student's $t$ test). **d** Protein was extracted from CHP134 and SK-N-AS cells stably transfected with an empty vector or JMJD6 expression construct, followed by immunoblot analysis. **e** Protein extracted from CHP134 cells was subjected to IP overnight with 5 μg of control IgG or anti-N-Myc Ab, followed by immunoblot analysis with anti-JMJD6 or anti-N-Myc Ab. **f** Immobilized GST-N-Myc protein fragments were incubated with equal amounts of nuclear protein prepared from HEK-293T cells transiently transfected with the recombinant vector pCMV14-JMJD6_3 × Flag. Pulled-down complexes were probed with a monoclonal anti-Flag antibody, and Ponceau staining detected by ChemiDoc MP was used as loading controls. **g** DNA-protein complex was extracted from CHP134 cells for ChIP sequencing with anti-JMJD6 and anti-N-Myc antibodies. Numbers inside the Venn diagram indicated the numbers of peaks at super-enhancers, typical enhancers, and promoters, which were bound by JMJD6 or N-Myc, and the overlap between them. Source data are provided as a Source Data file

N-Myc and c-Myc mRNA, and protein expression in CHP134 and SK-N-AS cells (Fig. 3c). Consistent with these data, transfection with JMJD6 siRNA-1 or siRNA-2 reduced N-Myc and c-Myc mRNA and protein expression (Supplementary Fig. 3e, f), and transfection with a JMJD6 open reading frame (ORF) expression construct led to E2F2, N-Myc, and c-Myc upregulation (Fig. 3d), in CHP134 or SK-N-AS cells. While JMJD6 has been shown to activate gene transcription through forming a protein complex with BRD4[7] and BRD4 is well known to regulate N-Myc and c-Myc gene transcription[25,26], treatment with the small molecule BRD4 inhibitor OTX015 also considerably reduced E2F2 expression in CHP134 and SK-N-AS cells (Supplementary Fig. 3g). In addition, c-Myc is known to directly induce *E2F2* gene transcription by binding to Myc-responsive element E-boxes at the *E2F2* gene promoter[27]. In CHP134 and SK-N-AS cells, knocking down N-Myc or c-Myc with two independent siRNAs significantly reduced E2F2 mRNA and protein expression (Supplementary Fig. 4a, b). Conversely, upregulation of N-Myc by DOX withdrawal in SHEP Tet/21N cells upregulated E2F2 mRNA and protein expression (Supplementary Fig. 4c, d).

Since both JMJD6 and N-Myc are known to activate gene transcription[7,14], we examined whether JMJD6 formed a protein complex with N-Myc in cells and whether JMJD6 directly binds to N-Myc. Protein co-immunoprecipitation and immunoblot analyses showed that an anti-N-Myc antibody efficiently immunoprecipitated both JMJD6 and N-Myc proteins in CHP134 cells (Fig. 3e). Different N-Myc protein fragments (1–86, 82–254, 249–358, 336–400, and 400–464 amino acids) were then cloned into the pGEX-2T construct, in frame with N-terminal GST. GST pull-down assays showed that JMJD6 protein was bound to the N-Myc 82–254 amino acid fragment (Fig. 3f), demonstrating that JMJD6 protein directly binds to the Myc Box II region of N-Myc protein.

To further demonstrate the JMJD6-N-Myc protein complex, we performed ChIP-Seq with anti-JMJD6 and anti-N-Myc antibodies in CHP134 cells. Bioinformatics analysis showed that both JMJD6 and N-Myc protein were bound to super-enhancers, typical enhancers, as well as promoters. Importantly, the majority of super-enhancers and typical enhancers bound by JMJD6 were also bound by N-Myc, and ~47% of promoters bound by JMJD6 were also bound by N-Myc (Fig. 3g, Supplementary Data 3–5). Taken together, our data suggest that JMJD6 forms a protein complex with BRD4 and N-Myc, and therefore regulates E2F2 and Myc expression as well as E2F and Myc target gene expression.

**JMJD6 induces neuroblastoma cell proliferation and survival.** As JMJD6 regulates Myc and E2F2 expression, we examined whether JMJD6 modulates neuroblastoma cell proliferation, survival, and clonogenic capacity. CHP134 and SK-N-AS cells were transfected with control siRNA, JMJD6 siRNA-1 or JMJD6 siRNA-2. Alamar blue assays showed that knocking down JMJD6 reduced the number of viable CHP134 and SK-N-AS cells (Fig. 4a). As the JMJD6 siRNAs targeted the 5′-untranslated region of JMJD6, CHP134 and SK-N-AS cells stably transfected with an empty vector or JMJD6 ORF expression construct were transfected with control siRNA, JMJD6 siRNA-1 or JMJD6 siRNA-2. Alamar blue assays showed that forced overexpression of JMJD6 largely reversed the effect of JMJD6 siRNAs in reducing the number of viable neuroblastoma cells (Fig. 4b). Similarly, treatment with DOX resulted in a significant decrease in the number of viable DOX-inducible JMJD6 shRNA-1 and JMJD6 shRNA-2, but not DOX-inducible control shRNA, CHP134, and SK-N-AS cells (Supplementary Fig. 5a, b). Cell cycle analysis showed that treatment with DOX had no effect on the percentage of DOX-inducible control shRNA CHP134 and SK-N-AS cells at each phase of the cell cycle, but significantly increased the percentage of DOX-inducible JMJD6 shRNA-1 and JMJD6 shRNA-2 CHP134 and SK-N-AS cells at the sub-G1 phase and decreased the percentage of cells at the S phase (Fig. 4c and Supplementary Fig. 5c, d).

Colony formation assays were also performed. As shown in Fig. 4d, treatment with DOX did not have an effect on clonogenic capacity in DOX-inducible control shRNA, but considerably diminished clonogenic capacity in DOX-inducible JMJD6 shRNA-1 and JMJD6 shRNA-2, CHP134 and SK-N-AS cells. Taken together, the data suggest that JMJD6 is required for neuroblastoma cell proliferation, survival, and tumorigenic capacity.

**JMJD6 is important for neuroblastoma progression in vivo.** To determine whether JMJD6 is important for E2F2 and Myc expression and neuroblastoma progression in vivo, DOX-inducible JMJD6 shRNA-2 SK-N-AS cells were xenografted into mice, and the mice were fed food with or without DOX.

The DOX treatment group displayed considerably slower tumor growth in comparison to the control treatment group (Fig. 5a). Survival curve analysis showed that DOX treatment, compared with control treatment, significantly increased the probability of survival in mice xenografted with DOX-inducible JMJD6 shRNA-2 cells by twofold (Fig. 5b). Immunoblot analysis showed that DOX treatment, compared with vehicle control treatment, significantly decreased JMJD6 protein expression in DOX-inducible JMJD6 shRNA SK-N-AS cell tumors (Fig. 5c, d). Protein expression of the JMJD6 targets, E2F2 and c-Myc, was also reduced in the DOX treatment group, compared with the vehicle control treatment group (Fig. 5c, d). The data

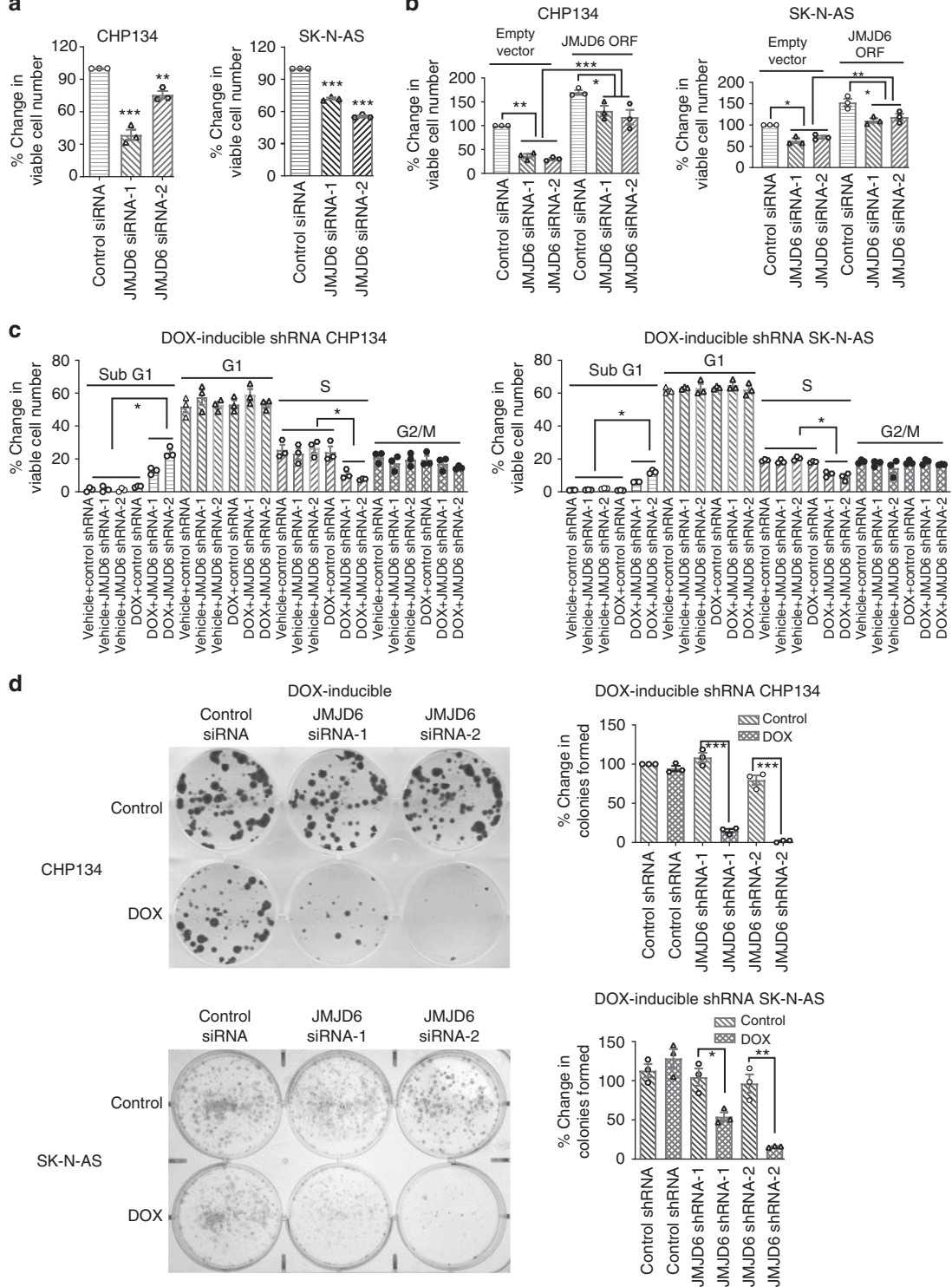

**Fig. 4** JMJD6 is important for neuroblastoma cell proliferation, survival, and colony formation. **a** CHP134 and SK-N-AS cells were transfected with control siRNA, JMJD6 siRNA-1 or JMJD6 siRNA-2 for 96 h, followed by Alamar blue assays. Error bars represent normalized standard errors from three independent experiments (**$p < 0.01$, ***$p < 0.001$, one-way ANOVA). **b** CHP134 and SK-N-AS cells stably transfected with an empty vector or JMJD6 open reading frame (ORF) expression construct were transfected with control siRNA, JMJD6 siRNA-1 or JMJD6 siRNA-2, followed by Alamar blue assays. Error bars represent normalized standard errors from three independent experiments (*$p < 0.05$, **$p < 0.01$, ***$p < 0.001$, one-way ANOVA). **c** DOX-inducible control shRNA, JMJD6 shRNA-1 or JMJD6 shRNA-2 CHP134, and SK-N-AS cells were treated with control or DOX for 72 h, followed by staining with propidium iodide and flow cytometry analysis of the cell cycle. The percentage of cells at each phase of the cell cycle was calculated. Error bars represent standard errors from three independent experiments (*$p < 0.05$, two-tailed unpaired Student's $t$ test). **d** DOX-inducible control shRNA, JMJD6 shRNA-1 or JMJD6 shRNA-2 CHP134, and SK-N-AS cells were treated with vehicle control or DOX for 14 days. Tumor cells were stained with crystal violet and the numbers of colonies were counted. Error bars represent normalized standard errors from three independent experiments (*$p < 0.05$, **$p < 0.01$, ***$p < 0.001$, two-tailed unpaired Student's $t$ test). Source data are provided as a Source Data file

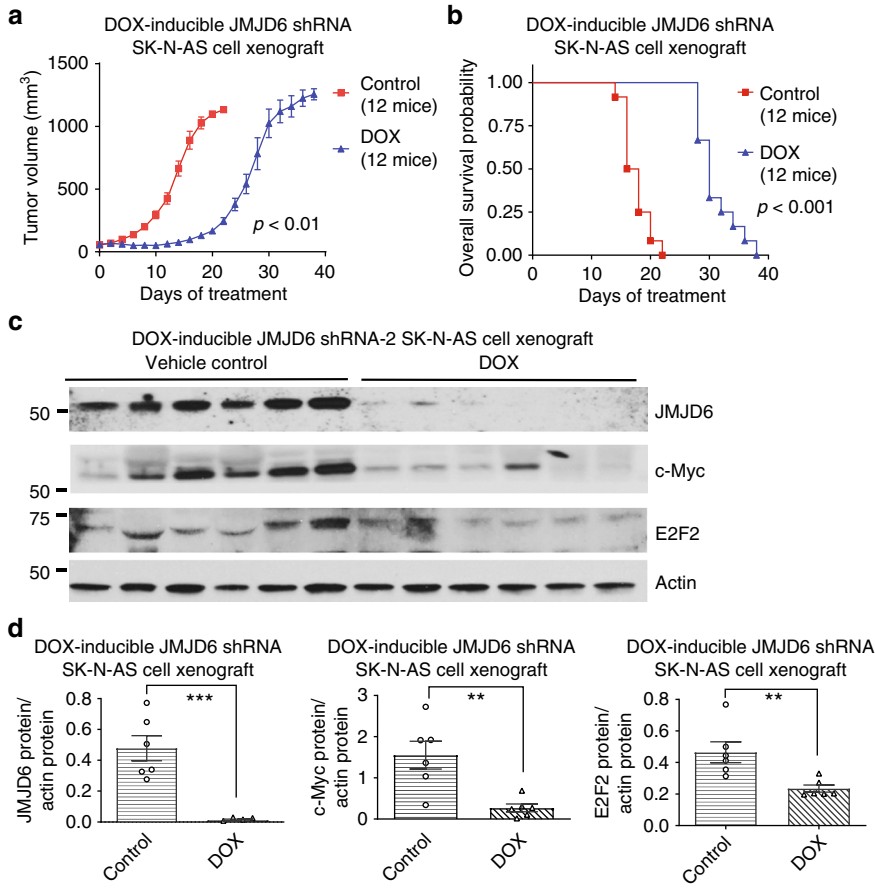

**Fig. 5** JMJD6 is important for E2F2 and Myc expression and tumor progression in vivo. **a** DOX-inducible JMJD6 shRNA-2 SK-N-AS cells were xenografted into nude mice. When tumors reached 0.05 cm$^3$ in volume, the mice were divided into two groups, and fed with food with or without 600 mg/kg of DOX. Tumor growth was monitored every other day, and the mice were culled when tumor reached 1 cm$^3$. Error bars represent standard errors (**$p < 0.01$, two-tailed paired Student's $t$ test). **b** Survival curve analysis showed the probability of mouse overall survival. Two-sided log-rank test was used to generate the $p$-value. **c**, **d** Protein was extracted from the tumors from the mice, and subjected to immunoblot analysis of JMJD6, c-Myc, and E2F2 protein expression (**c**), followed by quantification against the house-keeping actin using Quantity One software (**d**). Error bars represent standard errors (**$p < 0.01$, ***$p < 0.001$, two-tailed unpaired Student's $t$ test). Source data are provided as a Source Data file

demonstrate that JMJD6 is important for E2F2 and Myc expression, neuroblastoma tumor progression, and mouse survival in vivo.

**THZ1 and panobinostat synergistically reduce JMJD6 expression.** Critical oncogenes are characterized by association with transcriptional super-enhancers that induce massive oncogene overexpression[10,14]. We performed ChIP-Seq with anti-acetylated histone H3 lysine 27 (H3K27ac), anti-trimethyl H3K4 (H3K4me3), and anti-monomethyl H3K4 (H3K4me) antibodies in CHP134 cells. We also analyzed the published anti-H3K27ac antibody ChIP-Seq data from SK-N-AS cells, which were deposited at Gene Expression Omnibus website (Series GSE90683)[28]. The ChIP-Seq data revealed that the *MXRA7/JMJD6* gene locus was associated with H3K27 acetylation and H3K4 monomethylation but not with H3K4 tri-methylation (Fig. 6a, b), markers for transcriptional super-enhancers. ChIP PCR showed that CDK7 bound more abundantly to the *MXRA7/JMJD6* gene super-enhancer region than the *JMJD6* gene promoter region (Supplementary Fig. 6a).

We then investigated the effect of the CDK7/super-enhancer inhibitor THZ1 on *JMJD6* and *MXRA7* gene expression. RT-PCR and immunoblot analyses showed that treatment with THZ1 reduced JMJD6 and N-Myc expression in CHP134 cells, and JMJD6 and c-Myc expression in SK-N-AS cells (Fig. 6c), but showed no effect on MXRA7 expression (Supplementary Fig. 6b).

The data suggest that the *MXRA7/JMJD6* gene super-enhancers activate the transcription of the *JMJD6* but not *MXRA7* gene.

Since BET bromodomain inhibitors and HDAC inhibitors are known to synergistically reduce oncogene expression[29], we examined whether combination therapy with THZ1 and the HDAC inhibitor panobinostat synergistically reduced JMJD6, Myc and their target E2F2 expression. RT-PCR and immunoblot analyses showed that JMJD6, N-Myc, c-Myc, and E2F2 mRNA and protein were all more significantly reduced by the THZ1 and panobinostat combination treatment, compared with single-agent treatment (Fig. 6d and Supplementary Fig. 7a, b). Thus, the data suggest that the *JMJD6* gene is associated with transcriptional super-enhancers, and that THZ1 and panobinostat combination therapy synergistically reduces JMJD6, N-Myc, c-Myc, and E2F2 expression.

**THZ1 and panobinostat synergistically suppress neuroblastoma.** We next examined the synergistic effects of THZ1 and panobinostat in inhibiting neuroblastoma cell proliferation and survival. Alamar blue assays showed that treatment with THZ1 in combination with panobinostat for 72 h significantly decreased the numbers of viable CHP134 and SK-N-AS cells, compared with treatment with THZ1 or panobinostat alone (Fig. 7a). Isobologram analysis confirmed that the combination was indeed synergistic, as the combination indexes were well below 1.0 (Fig. 7b). We next examined whether JMJD6 was required for the anticancer effects of

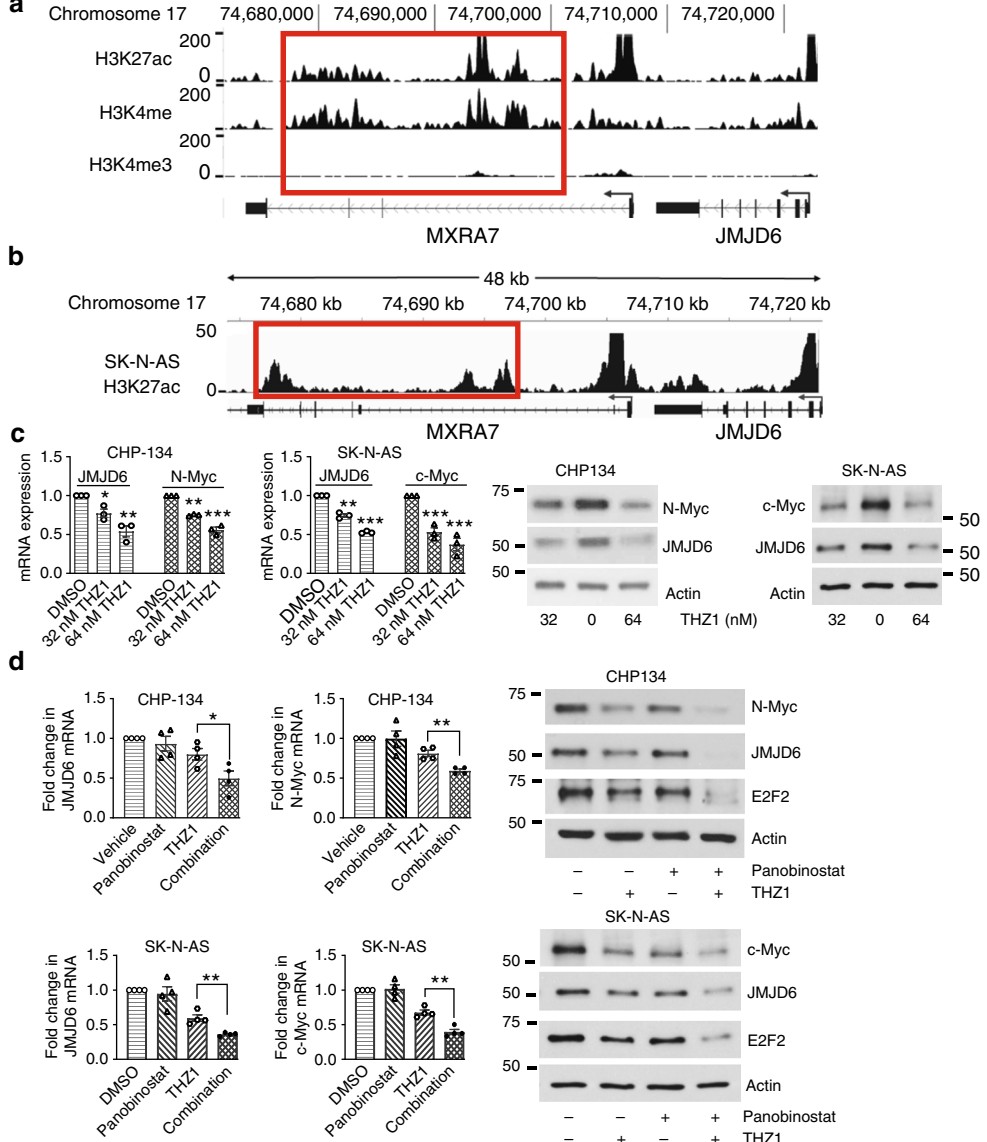

**Fig. 6** THZ1 and panobinostat synergistically reduce JMJD6, E2F2, and Myc expression. **a** ChIP-Seq was performed with control IgG, mono-methyl H3K4 (H3K4me), trimethyl H3K4 (H3K4me3), or acetyl H3K27 (H3K27ac) antibodies in CHP134 cells. H3K4me, H3K4me3, and H3K27ac occupancy at the *MXRA7* and *JMJD6* gene loci was analyzed. The *x*-axis indicates genomic position and the *y*-axis depicts the ChIP-seq signal from each histone mark, in arbitrary units normalized against input control, and super enhancer regions were boxed in red. **b** ChIP-Seq profiling for H3K27ac occupancy at the *MXRA7* and *JMJD6* gene loci in SK-N-AS cells was analyzed, using the published Series GSE90683 dataset. **c** CHP134 and SK-N-AS cells were treated with vehicle control, 32 or 64 nM THZ1. RNA and protein were extracted from the cells 24 and 48 h later, respectively, and subjected to RT-PCR and immunoblot analysis of JMJD6, N-Myc, and c-Myc mRNA and protein expression. Error bars represent normalized standard errors from three independent experiments (*$p < 0.05$, **$p < 0.01$, ***$p < 0.001$, two-way ANOVA). **d** CHP134 and SK-N-AS cells were treated with vehicle control, 32 nM THZ1, 10 nM panobinostat, or combination of THZ1 and panobinostat for 48 h, followed by RT-PCR and immunoblot analyses of JMJD6, N-Myc, and c-Myc mRNA and protein expression. Error bars represent normalized standard errors from four independent experiments (*$p < 0.05$, **$p < 0.01$, two-tailed unpaired Student's *t* test). Source data are provided as a Source Data file

this combination therapy, by treating CHP134 and SK-N-AS cells stably transfected with an empty vector or JMJD6 ORF expression construct with THZ1 and panobinostat, either alone or in combination. Alamar blue assays showed that overexpression of JMJD6 partially reversed the reduction in the number of viable neuroblastoma cells after treatment with THZ1 and panobinostat combination therapy (Fig. 7c). In addition, cell cycle analysis revealed that THZ1 and panobinostat synergistically increased the percentage of cells at the sub-G1 phase in both CHP134 and SK-N-AS cells (Fig. 7d and Supplementary Fig. 7c, d).

We lastly xenografted SK-N-AS neuroblastoma cells stably transfected with an empty vector or JMJD6 ORF expression construct into nude mice, and treated these mice with vehicle control, THZ1, panobinostat or combination, when tumors reached 0.05 cm³. As shown in Fig. 7e, treatment with THZ1 or panobinostat alone reduced tumor growth in mice xenografted with empty vector SK-N-AS cells and treatment with panobinostat alone reduced tumor growth in mice xenografted with SK-N-AS cells transfected with JMJD6 ORF. Importantly, THZ1 and panobinostat combination therapy synergistically reduced tumor

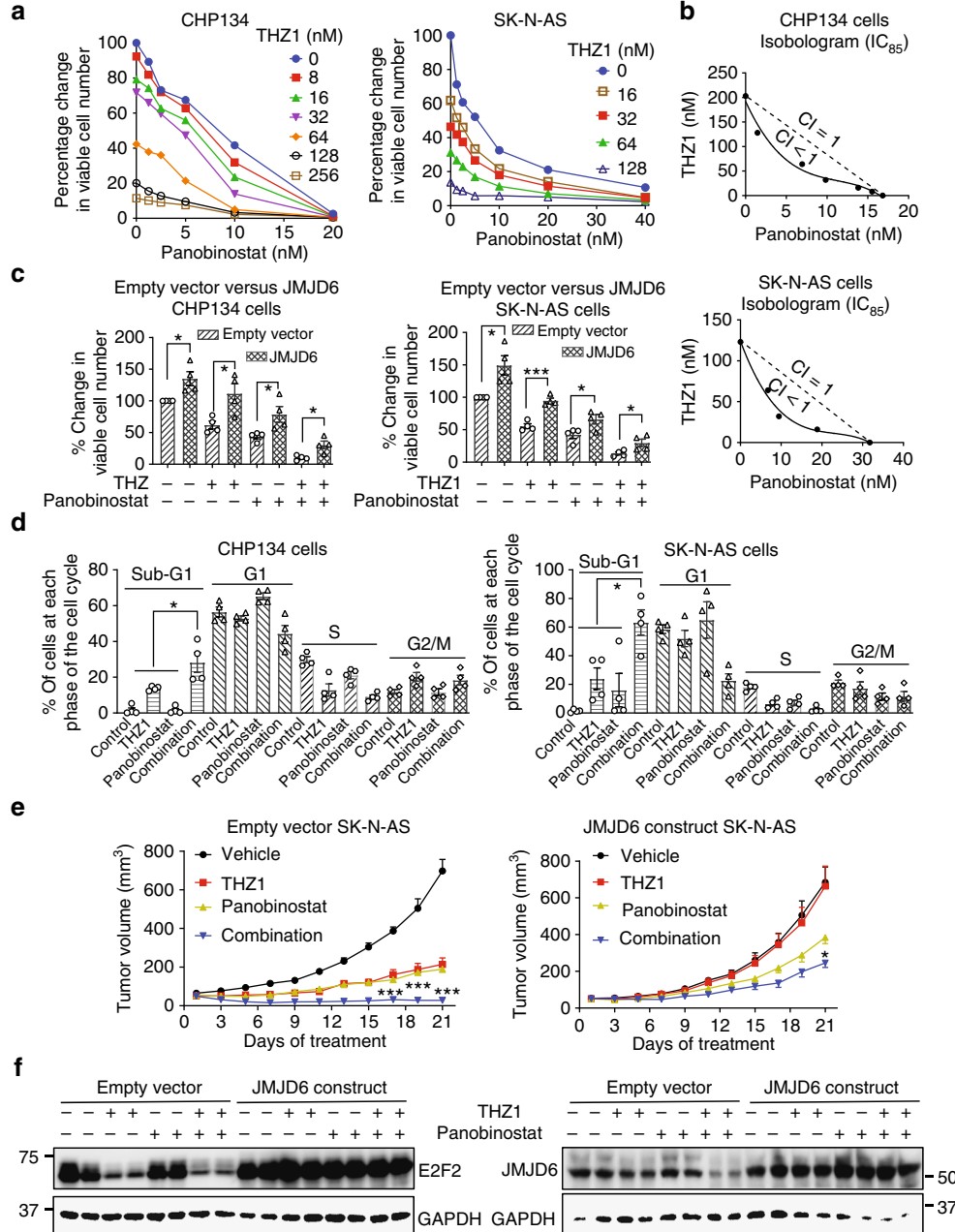

**Fig. 7** THZ1 and panobinostat synergistically induce tumor regression by reducing JMJD6. **a**, **b** CHP134 and SK-N-AS cells were treated with vehicle control, THZ1, panobinostat, or their combination for 72 h, followed by Alamar blue assays (**a**). Error bars represent normalized standard errors from three independent experiments. Isobolograms showed the actual concentrations of THZ1 and panobinostat required to achieve inhibition of cell viability by 85% (IC$_{85}$, solid line), as compared with additivity [dotted line, combination index (CI) = 1] (**b**). **c** CHP134 and SK-N-AS cells stably transfected with an empty vector or JMJD6 expression construct were treated with vehicle control, 32 nM THZ1, 10 nM panobinostat, or their combination for 72 h, followed by Alamar blue assays. Error bars represent normalized standard errors from four independent experiments (*$p < 0.05$, ***$p < 0.001$, two-tailed unpaired Student's $t$ test). **d** CHP134 and SK-N-AS cells were treated with vehicle control, 32 nM THZ1, 10 nM panobinostat, or their combination for 72 h, followed by staining with propidium iodide and flow cytometry analysis of the cell cycle. Error bars represent standard errors from four independent experiments (*$p < 0.05$, two-tailed unpaired Student's $t$ test). **e** Seventy-two nude mice were xenografted with SK-N-AS cells stably transfected with an empty vector or JMJD6 ORF expression construct. When tumors reached 0.05 mm$^3$, the mice were divided into four subgroups of nice mice each and treated intraperitoneally with control solvent, THZ1 (10 mg/kg body weight/twice a day), panobinostat (7.5 mg/kg body weight/3 days), or THZ1 plus panobinostat for 21 days. Tumor mass was measured every other day. Error bars represent standard errors (*$p < 0.05$, ***$p < 0.001$, two-way ANOVA). **f** Protein was extracted from the tumor tissues, and subjected to immunoblot analysis of JMJD6 and E2F2 protein expression. Source data are provided as a Source Data file

progression and resulted in tumor regression in mice xenografted with empty vector SK-N-AS cells, but only moderately suppressed tumor progression in mice xenografted with SK-N-AS cells transfected with JMJD6 ORF (Fig. 7e). Immunoblot analysis of

tumor tissues from the mice showed that THZ1 and panobinostat cooperatively reduced JMJD6 and E2F2 protein expression only in mice xenografted with empty vector SK-N-AS cells, and that transfection with the JMJD6 ORF expression construct led to

considerable JMJD6 and E2F2 protein overexpression and nonresponse to THZ1 and panobinostat therapy (Fig. 7f). Our data therefore demonstrate that THZ1 and panobinostat combination therapy synergistically blocks neuroblastoma progression and induces tumor regression by blocking JMJD6 expression.

## Discussion

Unlike adult cancers, neuroblastoma is characterized by frequent chromosome copy number variations, such as chromosome 17q21-ter gain[1]. However, it is not clear which gene or genes at 17q21-ter plays key oncogenic roles. In this study, by analyzing matched human neuroblastoma tissue array-CGH and gene expression datasets, we have confirmed that 17q21-ter/*JMJD6* gene is gained in the majority of *MYCN*-amplified and non-amplified human neuroblastoma tissues, and that 17q segmental gain leads to higher 17q copy number and higher *JMJD6* gene expression than 17q numerical gain and 17q no gain. Importantly, chromosome 17q segmental gain and high levels of JMJD6, but not 17q numerical gain or high levels of *MXRA7*, the gene immediately upstream of *JMJD6* at chromosome 17qter, in neuroblastoma tissues correlate with poor patient prognosis. In addition, our data demonstrate that JMJD6 overexpression in neuroblastoma tissues predicts poor patient prognosis, independently of other prognostic markers such as *MYCN* amplification status, age at the time of diagnosis, and disease stage.

We have found positive correlation between JMJD6 and N-Myc/c-Myc expression in human neuroblastoma tissues. We have also identified Myc-responsive element E-boxes at the *JMJD6* gene core promoter, confirmed N-Myc and c-Myc protein binding to the *JMJD6* gene core promoter, and demonstrated upregulation of JMJD6 by N-Myc and c-Myc. As N-Myc and c-Myc are well known to induce gene transcription by binding to Myc-responsive element E-boxes at target gene promoters[23,30], our data suggest that N-Myc and c-Myc overexpression in human neuroblastomas further increases JMJD6 expression, even in *JMJD6* gene-gained cells.

In this study, our genome-wide differential gene expression and GSEA analyses reveal that knocking down JMJD6 considerably reduces the expression of E2F target genes and the *E2F2* gene. ChIP-Seq data and GSEA analysis show that knocking down JMJD6 significantly reduces RNA Pol II binding to E2F and Myc target gene promoters. Importantly, we have confirmed that JMJD6 protein forms a complex with BRD4 protein in neuroblastoma cells, that JMJD6 protein directly binds to N-Myc protein at the Myc Box II region, and that the majority of super-enhancers, typical enhancers, and promoters bound by JMJD6 are also bound by N-Myc. In the literature, N-Myc regulates gene transcription mainly through binding to target gene promoters and enhancers[14,30], and JMJD6 activates gene transcription by interacting with MED12 in the mediator complex, interacting with BRD4 at antipause enhancers and binding to gene promoters[7,31–33]. Our data therefore suggest that JMJD6 activates gene transcription partly through binding to N-Myc, and that the interaction between JMJD6 and N-Myc is likely to be important for efficient gene transcription.

While E2F and Myc are well-known tumorigenic factors, we have confirmed that knocking down JMJD6 considerably reduces neuroblastoma cell proliferation, induces apoptosis, and dramatically reduces clonogenic capacity. Consistent with these in vitro data, knocking down JMJD6 reduces E2F2 and Myc expression, suppresses tumor progression, and improves survival in mice. The data suggest that high levels of JMJD6 expression due to chromosome 17q21-ter segmental gain contributes to neuroblastoma pathogenesis.

Currently no small molecule JMJD6 inhibitors are available. We have found that the *JMJD6* gene is associated with transcriptional super-enhancers in *JMJD6* gene-gained neuroblastoma cells, and that treatment with the CDK7 inhibitor THZ1, which specifically targets super-enhancer associated oncogenes[12–14], significantly reduces JMJD6, N-Myc, and c-Myc expression. While HDAC inhibitors are among the anticancer agents that exert the best synergistic anticancer effects with BET bromodomain inhibitors, through synergistically reducing oncogene expression[29], we have confirmed that combination therapy with THZ1 and the HDAC inhibitor panobinostat synergistically reduces JMJD6, N-Myc, c-Myc, and E2F2 expression and neuroblastoma cell proliferation, and synergistically induces neuroblastoma cell death, which can be partly blocked by JMJD6 overexpression. In neuroblastoma-bearing mice, THZ1 and panobinostat cooperatively reduce JMJD6 and E2F2 expression, blocks tumor progression, and results in tumor regression, and that forced JMJD6 overexpression in neuroblastoma cells significantly blocks the in vivo anticancer effects of THZ1 and panobinostat combination therapy. Our data therefore confirm the critical role of JMJD6 in neuroblastoma tumorigenesis. As the CDK7 inhibitor SY-1365 is currently in clinical trials in cancer patients [https://clinicaltrials.gov/ct2/show/NCT03134638] and the HDAC inhibitor panobinostat is currently in clinical practice, our findings identify CDK7 inhibitor and HDAC inhibitor combination therapy as a treatment strategy for neuroblastoma.

## Methods

**Cell culture**. SK-N-AS, SHEP Tet/21N, and 293T embryonic kidney cells were cultured in DMEM supplemented with 10% fetal calf serum. CHP134 cells were cultured in Roswell Park Memorial Institute Medium 1640 with 10% fetal calf serum and 1% l-glutamine. A total of 293T cells were obtained from the American Type Culture Collection 20 years ago. CHP134 and SK-N-AS cells were obtained from the European Collection of Cell Cultures and Sigma Aldrich in 2010. Cells were confirmed to be free from mycoplasma contamination, and cell line identity was verified in 2014–2018 by small tandem repeat profiling conducted at the Garvan Institute of Medical Research or Cellbank Australia.

**DOX-inducible control shRNA or JMJD6 shRNA cell lines**. The lentiviral DOX-inducible GFP-IRES-shRNA FH1tUTG construct was provided by Dr. Marco Herold[24] and used to generate control shRNA and JMJD6 shRNA expression constructs. Control shRNA, JMJD6 shRNA-1, and JMJD6 shRNA-2 target sequences were GCACTACCAGAGCTAACTCAGATAGTACT, CGAAGCTATT ACCTGGTTTAA, and ATGGACTCTGGAGCGCCTAAA, respectively. Sense and antisense control shRNA, JMJD6 shRNA-1, and JMJD6 shRNA-2 oligoes were cloned into the DOX-inducible GFP-IRES-shRNA FH1tUTG construct. The DOX-inducible GFP-IRES-control shRNA, JMJD6 shRNA-1 or JMJD6 shRNA-2 construct was transfected into 293T cells. Viral media were collected to infect CHP134 and SK-N-AS neuroblastoma cells with polybrene (Santa Cruz Biotechnology, Santa Cruz, CA) for 72 h. Cells with high GFP protein expression were selected by fluorescence-activated cell sorting with BD FACS Jazz™ II Cell Sorter (BD Biosciences, Franklin Lakes, NJ). For inducing shRNA expression in vitro, the cells were treated with 2 µg/ml DOX every 24 h.

**siRNA transfection**. Target sequences of siRNAs were: 5′-AATGTGAATA GTGCCAAGAAA-3′ (siRNA-1) and 5′-TGACAGAGCCCAAGAATGATT-3′ (siRNA-2) (Qiagen, Hamburg, Germany) for JMJD6; 5′-CGTGCCGGAGTTGGT AAAGAA-3′ (siRNA-1) and 5′-TCCAGCGAGCTGATCCTCAAA-3′ (siRNA-2) (Qiagen) for N-Myc; and 5′-CGGUGCAGCCGUAUUUCUATT-3′ (siRNA-1) and 5′-GGAACUAUGACCUCGACUATT-3′ (siRNA-2) for c-Myc. Negative control siRNAs did not target any human genes (All Stars Negative Control siRNA, Qiagen).

Neuroblastoma cells were transfected with siRNAs while plated onto 24- or 6-well plates or T25 flasks using Lipofectamine 2000 (Invitrogen, Carlsbad, CA) according to the manufacturer's instructions. Forty-eight hours later, RNA or protein was harvested for analyses.

**JMJD6 overexpressing neuroblastoma cell lines**. EF1a-empty vector-IRES2-mCherry-IRES-Puromycin-Lv224 and EF1a-JMJD6-IRES2-mCherry-IRES-Puromycin-Lv224 expression constructs were purchased from GeneCopoeia™ (Catalog number EX-E2359-Lv224, GeneCopoeia™, Rockville, MD). The constructs were transfected into 293T cells using Lipofectamine 2000 (Invitrogen, Carlsbad, CA). Viral media were collected to infect CHP134 and SK-N-AS neuroblastoma

cells with polybrene (Santa Cruz Biotechnology, Santa Cruz, CA) for 72 h. Cells with high mCherry protein expression were selected by fluorescence-activated cell sorting with BD FACS Jazz™ II Cell Sorter (BD Bioscience).

**Real-time reverse transcription PCR**. RNA was extracted from tumor cells with RNeasy Plus Mini Kit (Qiagen), followed by quantification with a Nanodrop spectrophotometer (Thermo Fisher Scientific, Waltham, MA), according to the manufacturer's instructions. cDNAs were synthesized with random hexonucleotide primers and Moloney murine leukemia virus reverse transcriptase (Invitrogen). RT-PCR was performed with Power SYBR Green Master Mix (Invitrogen) as the fluorescent dye and gene specific primers using Applied Biosystems 7900 (Applied Biosystems, Grand Island, NY). No template controls were employed for detecting nonspecific amplification. The sequences of RT-PCR primers were: 5′-CGACCAC AAGGCCCTCAGTA-3′ (forward) and 5′-CAGCCTTGGTGTTGGAGGAG-3′ (reverse) for N-Myc; 5′-GGATTTTTTTCGGGGTAGTGGAA-3′ (forward) and 5′-TTCCTGTTGGTGAAGCTAACGTT-3′ (reverse) for c-Myc; 5′- GATCCAGAC TCGCACTGGAC-3′ (forward) and 5′- CCCGAGGTCAGAGGTTTGTT-3′ (reverse) for JMJD6; 5′-GGCCAAGAACAACATCCAGT-3′ (forward) and 5′-CGTGTTCATCAGCTCCTTCA-3′(reverse) for E2F2; and 5′-AGGCCAACCGCG AGAAG-3′ (forward) and 5′-ACAGCCTGGATAGCAACGTACA-3′ (reverse) for Actin. The primers were synthesized by Sigma (Sigma, Sydney, Australia). The comparative threshold cycle ($\triangle\triangle$Ct) method was employed to quantify fold changes in the expression of target genes[34], relative to the housekeeping gene actin.

**Immunoblot**. Protein was extracted from tumor cells with radio-immunoprecipitation assay buffer (150 mM NaCl, 1% NP-40, 0.5% sodium deoxycholate, 0.1% SDS, 50 mM Tris-Cl pH 7.5) containing protease inhibitors (Sigma, St Louis, MO, USA). Supernatant was collected after the samples were centrifugated at $13,000 \times g$ at 4 °C for 20 min. Protein concentrations in the supernatant samples were quantified with the Bicinchoninic Acid Assay kit (Pierce, Rockford, IL).

For immunoblotting, protein samples were loaded onto sodium dodecyl sulfate-polyacrylamide gels, followed by electrophoresis and transfer to nitrocellulose membranes. The membranes were blocked of nonspecific antibody binding with 10% skim milk powder in phosphate-buffered saline, and probed with the following primary antibodies: mouse anti-E2F2 antibody (1:2000) (sc-633×), mouse anti-JMJD6 antibody (1:500) (sc-28348), mouse anti-N-Myc antibody (1:1000) (sc-53993), or rabbit anti-c-Myc antibody (1:500) (sc-764) (all from Santa Cruz Biotechnology). The membranes were then incubated with a horseradish peroxidase-conjugated goat anti-rabbit or goat anti-mouse antibody (1:10000) (both from Santa Cruz Biotechnology), and protein bands were visualized with SuperSignal (Pierce). The membranes were finally probed with a mouse anti-actin antibody (1:30000) (A3853, Sigma) as loading controls.

**Affymetrix microarray gene expression study**. DOX-inducible control shRNA, JMJD6 shRNA-1 or JMJD6 shRNA-2 CHP134 or SK-N-AS cells were treated with vehicle control or DOX for 40 h. RNA was extracted from the cells with RNeasy Mini Kit (Qiagen), and genome-wide differential gene expression was examined with Affymetrix microarrays using the Clariom S Human assay (Affymetrix, Santa Clara, CA)[29]. The microarray data were analyzed in R [http://www.r-project.org/] with bioconductor packages [http://www.bioconductor.org/] and normalized with the robust multiarray average algorithm[35], and differential expression analysis was performed using the Limma package[36]. Moderated t tests were performed with the Limma package and the false discovery rate (FDR) was controlled using the Benjamini and Hochberg method[37]. Functional enrichment of the differentially expressed genes was performed with GSEA[38]. The list of differentially expressed genes were ranked by the log of the fold change and then compared with known gene sets in the curated database Molecular Signature Database (MSigDB), in particular, the C3: Motif gene sets database of genes sets[39]. A gene set was considered to be significantly enriched if its FDR q-value was below 0.25.

**ChIP assays**. For analyzing protein binding to gene promoters and enhancers, cells were incubated with formaldehyde to cross-link protein and DNA. After cell lysates were sonicated, a mouse anti-N-Myc (sc-53993, Santa Cruz Biotechnology), rabbit anti-c-Myc (ab56, Abcam), mouse anti-RNA Pol II (664903, BioLegend, San Diego, CA), mouse anti-CDK7 (sc-7344, Santa Cruz Biotechnology) antibody, or control rabbit or mouse antibody were used to immunoprecipate the DNA-protein complex, followed by real-time PCR with primers targeting negative control regions, the core promoter regions of the JMJD6 or the MYCN genes or the JMJD6 super-enhancer[40]. The sequences of primers used were: 5′-GGCAATATCATCA CCCCCTA-3′ (forward) and 5′-GGCCAAGCTGATCTTGAACT-3′ (reverse) for the remote negative control region of the JMJD6 gene; 5′-AGGAACCCAGCCA AGTCAG-3′ (forward) and 5′-GCTTTCGATTCTCGGAGGAG-3′ (reverse) for the JMJD6 gene promoter containing canonical and noncanonical E-boxes; 5′-TTTGTTGAATGCGCAAGAG-3′ (forward) and 5′-GCCATCGTTATCCTCACC AC-3′ (reverse) for the JMJD6 gene exon 2 region; 5′-TCTAGGTCCCACTGGCA TTT-3′ (forward) and 5′-TGGTTTGAATGTGTCCTCCA-3′ (reverse) for the remote negative control region of the MYCN gene; 5′-TGGGTTAGAAGCATCGG TCT-3′ (forward) and 5′-CCTGGCAATTGCTTGTCATT-3′ (reverse) for the

MYCN gene promoter; 5′-CTCTCTCCCTCTCCCGACTT-3′ (forward) and 5′-TGGGGAGACACAGTGAACCAA-3′ (reverse) for the JMJD6 gene super-enhancer region; and 5′-CGGTGAGCCAAGATCATACC-3′ (forward) and 5′-CGCCTG ACGAGAAAAGACTC-3′ (reverse) for the JMJD6 gene intron 5 region. Fold enrichment of the gene promoter or super-enhancer region was calculated by dividing PCR products from samples immunoprecipitated by the experimental antibody by PCR products from samples immunoprecipitated by the control antibody.

**ChIP-Seq**. DOX-inducible control JMJD6 shRNA-2 CHP134 cells were treated with vehicle control or DOX for 40 h, followed by ChIP with a mouse anti-RNA Pol II antibody (664903, BioLegend) or control mouse IgG (sc-2025, Santa Cruz Biotech). In addition, ChIP was performed in CHP134 cells with a rabbit anti-H3K27ac antibody (ab4726, Abcam), rabbit anti-H3K4me antibody (ab106165, Abcam), rabbit anti-H3K4me3 antibody (ab213224, Abcam), mouse anti-N-Myc (sc-53993, Santa Cruz Biotechnology), rabbit anti-JMJD6 (ab64575, Abcam), or control rabbit IgG (sc-2027, Santa Cruz Biotech). After immunoprecipitation, DNA was purified and subjected to sequencing with Illumina HiSeq 2000 at Ramaciotti Center for Genomics, University of New South Wales, Sydney, Australia.

ChIP-Seq raw reads were aligned using the Burrows Wheeler Aligner with default parameters[41]. Files were further processed using SAMtools[42], BEDtools[43], and bedGraphToBigWig[44]. Peaks were called with MACS2 "callpeak" (version macs2 2.1.1.20160309) against input controls, with default parameters[45].

Gene promoters were selected as ±1 kb from the transcription start site of genes from the RefSeq (refFlat) database[46], downloaded from the University of California Santa Cruz Table Browser. Enhancer and super-enhancer regions were identified using the ROSE algorithm[10,11,48]. H3K27ac peaks within ±1000 bp of transcription start sites were subtracted, enhancers were sutured at 12,500 bp in distance for ranking, and a threshold for distinguishing enhancers and super-enhancers was determined according to the geometric inflection point[10,11,48]. For enhancer and super-enhancer gene associations, each gene was assigned a regulatory domain that extends ±1000 kb in both directions to the nearest gene's transcription start site using the Genomic Regions Enrichment of Annotations Tool[49]. The two nearest gene regions overlapping with a given enhancer or super-enhancer are listed alongside each region. Promoters, enhancers and super-enhancers bound by N-Myc or JMJD6 were selected if they overlapped with a ChIP-seq peak identified from all three replicates of either N-Myc or JMJD6, and only peaks with FDR q-value < 0.05 were retained for analysis.

To select genes with reduced RNA Pol II binding at their promoters after DOX treatment in DOX-inducible JMJD6 shRNA-2 CHP134 cells, the following steps were undertaken. Promoters with reduced RNA Pol II binding were selected as those that had a RNA Pol II ChIP-Seq peak called within the region in both replicates in vehicle control-treated samples, but no peak in either replicate in DOX-treated samples. The statistical difference between normalized numbers of RNA Pol II ChIP-Seq reads between vehicle control-treated samples and DOX-treated samples was determined separately for each replicate using a paired t test. The data points used to calculate statistical significance were "normalized ChIP-Seq reads per kilobase," determined using the following formula: total number of reads overlapping region of interest/size of region of interest (base pairs)/normalization factor × 1000, where normalization factor was equal to the number of ChIP-seq reads in the file divided by the average number of ChIP-seq reads across all files.

The resulting list of genes with promoters identified by the above analysis was used as input for the GSEA, performed with the "Investigate Gene Sets" tool available via the Broad Institute [http://software.broadinstitute.org/gsea/], and the MSigDB[38]. Genesets with FDR q-value < 0.05 and p < 0.05 in Hallmark Gene Sets were regarded as significantly regulated gene sets.

**Protein co-immunoprecipitation assays**. Protein was extracted from CHP134 cells and co-immunoprecipitation was carried out using a rabbit anti-BRD4 antibody (Bethyl Laboratories, Montgomery, TX), rabbit anti-JMJD6 antibody (Abcam ab135066) or control IgG as a negative control, followed by immunoblot analysis with anti-BRD4 and anti-JMJD6 antibodies. Protein co-immunoprecipitation was also carried out with an anti-N-Myc antibody (Santa Cruz Biotechnology B8.4.b) or control IgG as a negative control, followed by immunoblot analysis with the anti-JMJD6 (Abcam Ab64575) and an anti-N-Myc antibodies.

**GST pull-down assays**. Different N-Myc protein fragments (1–86, 82–254, 249–358, 336–400, and 400–464 aa) were cloned into the pGEX-2T construct, in frame with N-terminal GST[50]. The constructs were transformed into BL-21 E. Coli and IPTG was used for induction of T7-driven transcription (4 h at 30 °C). GST-N-Myc fragments were purified by using glutathione-agarose resin (Sigma - G4510). HEK-293T cells were transiently transfected with pCMV14-JMJD6_3 × Flag expression construct and nuclear protein from the cells was incubated with equal amount of different GST-N-Myc protein fragments immobilized onto glutathione agarose beads. Pulled-down complexes were analyzed by immunoblot with a monoclonal anti-Flag antibody (F1804, Sigma) and Ponceau staining detected by ChemiDoc MP (Bio-Rad) was used as loading controls.

**Alamar blue assays**. Alamar blue assays were performed as we described previously[40]. Briefly, cells in 96-well plates were transfected with various siRNAs, treated with vehicle control or DOX, or treated with vehicle control, THZ1 (CS-3168, Chemscene LLC, Monmouth Junction, NJ), and/or panobinostat (S1030, Selleckchem, Houston, TX). After 96 or 72 h, the cells were incubated with Alamar blue (Invitrogen) for the last 5 h, and plates were read on a microplate reader at 570/595 nm. Results were calculated according to the optical density absorbance units, and expressed as percentage changes in the number of viable cells, relative to the samples transfected with control siRNA or control shRNA or treated with vehicle control.

**Cell cycle analysis**. Neuroblastoma cells were treated with vehicle control, THZ1, panobinostat, or combination of THZ1 and panobinostat for 72 h. In separate experiments, DOX-inducible control shRNA, JMJD6 shRNA-1 or JMJD6 shRNA-2 neuroblastoma cells were treated with vehicle control or DOX for 72 h. Cells were then collected and resuspended in solution containing 2 μg/ml RNase (Sigma) and 50 μg/ml propidium iodide at $2 \times 10^6$ cells/ml (Sigma). The cells were thereafter examined by flow cytometry using FACS Calibur machine and FACS Diva software (BD Biosciences). The percentage of cells at every phase of the cell cycle was analyzed.

**In vivo mouse experiments**. Female Balb/c nude mice aged 5–6 weeks were anesthetized and subcutaneously injected with $8 \times 10^6$ DOX-inducible JMJD6 shRNA-2 SK-N-AS cells (50% serum-free cell culture medium/50% matrigel) into the right flank. Animals were excluded from experiments and analysis, if they did not develop tumors. When tumors reached 0.05 cm³, mice were randomly divided into two groups of 10–12 and fed with feed with or without 600 mg/kg DOX (Meat Free Rat and Mouse Finished Diet, Specialty Feeds Pty. Ltd., Glen Forrest, WA, Australia). Tumor development was monitored and tumor volume was calculated using (length × width × height)/2. All efforts were taken to measure tumor size as accurately as possible. When tumors reached 1.0 cm³, mice were culled and tumors were collected. This mouse experiment was approved by the Animal Care and Ethics Committee of UNSW Sydney, Australia, and animals' care was performed in agreement with institutional guidelines.

For the experimental therapy study, 72 male Balb/c nude mice aged 5–6 weeks were anesthetized and subcutaneously injected with $6 \times 10^6$ SK-N-AS cells transfected with empty vector or JMJD6 ORF expression construct (50% serum-free cell culture medium/50% matrigel) into the right flank. When tumors reached 0.05 cm³, mice in the empty vector SK-N-AS xenograft group and the JMJD6 ORF SK-N-AS xenograft group were each randomly divided into four subgroups of nice, and treated intraperitoneally with vehicle control, THZ1 at 10 mg/kg body weight twice every day, panobinostat at 7.5 mg/kg body weight once every three days, or combination of THZ1 and panobinostat. Tumor size was monitored once every other day. Mice were culled 21 days after the beginning of treatments, when tumors in vehicle control group approached 1.0 cm³. The mice were purchased from the Experimental Animal Center of Chinese Academy of Science (Shanghai, China). The experiment was approved by the Shanghai University of Traditional Chinese Medicine Committee on the Use of Live Animals for Teaching and Research, and animals' care was performed in accordance with the Guide for the Care and Use of Laboratory Animals published by the National Institutes of Health (publication No. SCXX(HU) 2007–0005).

**Analysis of 17q gain in human neuroblastoma tissues**. Array-CGH data from 209 human neuroblastoma tissues were generated previously and downloaded from the Gene Expression Omnibus website of the National Center for Biotechnology Information (SuperSeries GSE45480)[18]. JMJD6 expression in the 209 human neuroblastoma tissues were also published previously as a part of the Kocak dataset and downloaded from the Gene Expression Omnibus website (SuperSeries GSE45480)[18].

To create the map of chromosome 17 copy number gains, the whole chromosome was divided into bins of 100 kb and overlaps between intervals of copy number gains and these bins were counted. Only intervals exceeding a copy number of 2.5 were considered as gains. The frequency of gains among all 209 samples was plotted to show all bins across chromosome 17q.

**Statistical analysis**. For statistical analysis, experiments were performed at least three times. Data were analyzed with Graphpad Prism 6 and expressed as mean ± standard error. The variance was similar between the groups that were statistically compared. Differences were examined for significance with two-sided unpaired $t$ test for two groups or ANOVA among groups.

Correlation between JMJD6 expression with N-Myc and c-Myc expression, as well as correlation between JMJD6 expression and chromosome 17q gain, in human neuroblastoma tissues was examined using Pearson's correlation. Patient overall survival was specified as the time from diagnosis till death or till last contact if the patient did not die. Event-free survival was defined as the time from diagnosis until the first occurrence of disease relapse, progression, death, or until last contact if no event occurred. Survival analyses were performed according to the method of Kaplan and Meier using GraphPad Prism 6.0, and comparisons of survival curves were carried out using two-sided log-rank tests. Hazard ratios and survival probabilities were provided with 95% confidence intervals (CIs). Proportionality was verified by visual inspection of the plots of log(2 log(S(time))) versus log(time), which remained parallel[40]. A probability value of 0.05 or less was considered statistically significant.

**Reporting summary**. Further information on research design is available in the Nature Research Reporting Summary linked to this article.

## Data availability
The microarray data and the ChIP-Seq data have been deposited at the Gene Expression Omnibus website with series numbers of GSE112914, GSE113139, and GSE112919, respectively. All the other data supporting the findings of this study are available within the article and its supplementary information files and from the corresponding author upon reasonable request. A reporting summary for this article is available as a Supplementary Information file. The source data underlying Figs. 1b–f, 2a–c, e, 3a–f, 4a–d, 5a–d, 6c, d, 7a–f, and Table 1 and Supplementary Figs. 1a, b, 2a, b, 3a, d–g, 4a–d, 5a, b, 6a, b, and 7a, b are provided as a Source Data file. Array-CGH data from 209 human neuroblastoma tissues and gene expression in the 209 human neuroblastoma tissues were downloaded from the Gene Expression Omnibus website [https://www.ncbi.nlm.nih.gov/geo/query/acc.cgi?acc=GSE45480][18]. Gene expression in 88 and 476 human neuroblastoma samples and relevant patient prognosis information in the Versteeg and Oberthuer datasets were downloaded from the R2 platform [http://r2.amc.nl].

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

## Acknowledgements
We thank Dr. Marco Herold at Walter and Eliza Hall Institute of Medical Research, Melbourne, Australia, for providing the FH1tUTG construct. We thank Dr. Pauline Depuydt and Dr. Katleen De Preter for their kind help in analyzing chromosome 17q gain in human neuroblastoma tissues. Children's Cancer Institute Australia is affiliated with UNSW Australia and Sydney Children's Hospitals Network. The authors were supported by National Health and Medical Research Council Australia and Cancer Council NSW. P.Y.L. is a research fellow of Cancer Institute NSW, and R.C.P is supported by a National Health and Medical Research Council Early Career Fellowship.

## Author contributions
T.L. and Q.D. designed the study. M.W., Y.S., Z.X., G.M., R.C.P., C.B., J.L.B., C.M., N.H., A.E.T, X.C., Y.L., R.C., P.Y.L., C.C.J., Q.L., N.J., B.B.C., J.W.H.W. and H.X. performed the experiments, collected the data, and analyzed the results. R.E.G., G.P., H.X., M.F., S.H., J.Y.W., G.M.M., X.D.Z., M.D.N., M.H. and Q.L. provided conceptual advice. M.W., Q.D. and T.L. wrote the manuscript with contributions from the coauthors.

## Additional information

**Competing interests:** The authors declare no competing interests.

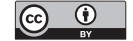

