## [Peer Review File · Nature Communications]

Reviewers' comments:

Reviewer #1 (Remarks to the Author):

This manuscript describes a possible oncogenic role of JMJD6 as a target of recurrent copy number gains affecting chromosome 17q in neuroblastoma. Authors conclude that:

- 1) Copy number gain of JMJD6 is frequently found in neuroblastoma tissues, particularly those carrying NMYC amplification.
- 2) Elevated expression of JMJD6 correlates with poor overall survival of neuroblastoma patients.
- 3) N-Myc and c-Myc upregulate JMJD6 expression through binding to the JMJD6 promoter.
- 4) JMJD6 is physically associated with BRD4 and N-Myc and required for the expression of E2F2, N-Myc, and c-Myc and also their transcriptional targets.
- 5) JMJD6 is required for neuroblastoma cell proliferation, survival and colony formation.
- 6) JMJD6 is required for expression of E2F2 and Myc and for progression of neuroblastoma in vivo.
- 7) THZ1, a CDK7 inhibitor, and panobinostat, a HDAC inhibitor, down-regulate JMJD6 and Myc expression, and finally
- 8) THZ1 and panobinostat synergistically induce neuroblastoma cell death in vitro and suppress tumor progression in vivo.

Although these conclusions are of potential interest, this reviewer raises several concerns, which should be properly addressed. Particularly confusing is that authors is trying to implicate JMJD6 expression in the pathogenic role of 17q gain, the most frequent genetic lesion in neuroblastoma without convincing data.

Major issues:

- 1) Authors implicitly suggest that upregulation of JMJD6 is by copy number gain and contributes to the pathogenesis of neuroblastoma. However, authors did not demonstrate any evidence that patients with 17q gains showed upregulation of JMJD6 expression, by comparing JMJD6 expression levels between samples with and without JMJD6 copy number gain, separately within NMYC amplification positive and negative cases.

- 2) Increased copy number of JMJD6 is found in 26% of all neuroblastoma cases and 63% of those with NMYC amplification according to their SNP array data. However, over all, 17q gain is found in the majority of neuroblastoma cases (~80-90% of all neuroblastoma), where the common copy number gain involves a more telomeric region than the JMJD6 locus. In other words, 17q genes located telomeric to JMJD5 certainly showed higher frequencies of copy number gain. Authors are requested to show a map of copy number gain in 17q in their samples and compare the frequency of copy number gain of other genes.
- 3) Authors claim a positive correlation between NMYC and JMJD6 expression levels (Fig. 1b). However, the correlation is primarily explained by higher JMJD6 expression in NMYC-amplified cases and within NMYC amplification-negative cases, almost no or very weak correlation, i.e., effect size, is observed. Similarly, despite low p-value for correlation between JMJD6 and c-Myc, the effect size is very low, if ever present.
- 4) If JMJD6 expression correlates with copy number, those cases showing higher JMJD6 expression are more likely to have larger deletions. Thus, the apparent poor prognosis of higher JMJD6 expression might be explained by larger 17q arm segment involved in these cases, but not the slightly higher JMJD6 expression.
- 5) What is the significance of the physical interaction between JMJD6 and BRD4? No data are shown supporting the role of this interaction. Also, the interaction should be supported by an opposite IP experiment, i.e., detection of BRD4 in IP with JMJD6.
- 6) Reduced E2F2 binding on its target promoters in JMJD6-directed siRNA-expressed cells should be demonstrate by CHIP seq for E2F2.
- 7) In Fig. 3e, N-myc is detected in control IgG lane, suspecting the precipitated N-Myc could be non-specific, but might not represent a true interaction.
- 8) siRNA may show negative effects on cell growth even when the results are obtained from 2 independent siRNAs. To exclude this, authors are requested to do re-expression analysis for the experiments in Fig. 4, in which JMJD6 should be overexpressed in siRNA treated cells. Authors overexpressed JMJD6 in Fig. 3d.
- 9) Supplementary Fig. 3b, the reduction of E2F2 expression does not correlate with the knockdown levels of N-Myc and c-Myc, raising a possibility that the reduction of E2F2 might have been off-target effect.
- 10) THZ1 and/or panobinostat should globally affect oncogenic superenhancers, not just that of JMJD6. Why authors could conclude that the effect of these compounds on cell growth and proliferation observed in vitro and in vivo was though disruption of the enhancer at JMJD6, not others? Authors are requested to revert the suppression of cell growth and tumor progression by re-expression of JMJD6.

Minor comments:

- 11) Fig. 5b: were there any censor cases? If not, Fig. 5b is a survival curve, not Kaplan-Meier estimate.
- 12) Discussion is too long and mostly repeats what were described in Results. Thus, it could be substantially shortened by focusing on the significance of the results.
- 13) JMJD6 may be important for expression of E2F2 and Myc proteins, but it is not demonstrated that it is required. So, the conclusions should be attenuated.

Reviewer #2 (Remarks to the Author):

In this investigation authors explore the functional role of histone demethylase JMJD6 in a paediatric cancer neuroblastoma. Here they show that JMJD6, which is a target of MYCN, is an oncogene and influences neuroblastoma through its interaction with MYCN and BRD4 and regulating target oncogenes such as E2F2 and MYCN. Based on their findings, authors propose use of THZ1 and panobinostat combination treatment in the treatment of neuroblastoma.

Overall it is an interesting manuscript. However, I find that the crucial findings in the manuscript are not well supported experimentally, in particular from high-throughput approaches. The data presentation is very poor. I have the following comments.

1. Authors state that there is a copy number gain of JMJD6. Are all these tumors have 17q gain? Are there any samples which have focal gain of JMJD6? It would good to show the extent of gain across the samples.
2. Does copy number gain correlate with RNA expression in those tumors? This is very important aspect and the data on this correlation analysis should be provided.
3. Authors propose a positive feedback loop between NMYC and JMJD6. However, there is a poor correlation value of 0.2 between NMYC and JMJD6 in tumor datasets. Authors should explain why such poor correlation exists?

4. Authors have used DOX inducible shRNA targeting JMJD6. However, experiments have been performed using only one shRNA. Authors should provide data from two or more shRNAs to rule out off-target effects. RNA-sequencing/microarray data should be provided from minimum of two shRNAs.

5. Microarray analysis lacks statistical details. Authors should describe what kind filtering criteria was adapted to identify JMJD6 target genes. In fact MYCN expression is upregulated in the microarray data, which is contrary to their Western and qRT-PCR data. The decrease in MYCN at the protein level after JMJD6 KD could be due to other type of regulation; transcriptional regulation might not be a predominant mechanism. Authors need to investigate this.

6. Authors have generated two stable cell lines of DOX inducible JMJD6 shRNA. However, microarray data has been provided from only one cell line. Since authors claim that JMJD6 is involved in the regulation of tumor promoting genes, such as E2F and MYCN, it would be interesting to check whether this is conserved across NB cell lines. Hence it would be good if the authors provide microarray/RNA-seq data from both CHP134 and SKNAS cell lines.

7. RNAP II ChIP-sequencing data lacks details and statistical information. XL sheets (Tables) lack information such as peak location, peak score, fold change and also the information of peaks existence in individual replicates. In particular I found the RNAP II peak consideration between control and DOX samples a bit strange (Genes with reduced RNA Pol II binding were therefore selected as those that had a RNA Pol II ChIP-seq peak called within the promoter in both replicates in vehicle control treated samples, but no peak in at least one replicate in DOX-treated samples). I am not sure whether it is correct way to consider the peaks differentially for control and DOX samples. Another important aspect is that is there any overlap between genes downregulated upon JMJD6 KD with loss of RNAP II peaks following JMJD6 loss? The presented data do not provide any insight how JMJD6 regulates gene transcription by MYCN interaction.

8. Since authors data claims that JMJD6 and NMYC functionally associated to regulate global gene expression, it would be interesting to see their colocalization across the genome (ChIP-seq) and also the functional overlap among the target genes of JMJD6 and NMYC to reinforce their common effects on global gene regulation in neuroblastoma. Does their colocalization specifically occurs at the super-enhancers associated with oncogenic drivers?

9. Figure 7f: I do not see any noticeable decrease of JMJD6, CMYC and E2F2 upon THZ1 in SKNAS xenografts while figure 6 does show effect on these proteins. Why this discrepancy? What is the IC50 concentration of THZ1 for CHP134 and SKNAS cell lines? E2F2 is shown as single band in

many immunoblots, is shown as two bands in THZ1 treatment in Figure 7f. This is also observed with JMJD6 and E2F2. There is an inconsistency in the presentation of the blots.

10. The use of CDK7 inhibitor in the current investigation is not clear. Moreover, the use of CDK7 inhibition for the MYCN driven tumor is not novel (<https://www.ncbi.nlm.nih.gov/pubmed/2541650>). CDK7 is bound to a subset of the super enhancers (<https://www.nature.com/articles/nature13393>) and regulate the expression of oncogenes such as RUNX1. Is the super enhancer that regulates JMJD6 bound by Cdk7?

Minor comments:

In Figure 2A, last panel C-MYC KD in SKNAS cells does not look biological although significance is provided.

Figure 2B: Authors should quantify Western bands

In Fig 6A: authors highlight enhancer-specific modification peaks in MXRA7 gene but not JMJD6? Does these hypothetical enhancers are specific to JMJD6 or MXRA7 or both?

Check the following sentence

“demethylating histone H4 arginine 3 (H4R3)”

Responses to Reviewers

Reviewer #1:

Major issues:

Question (1). Authors implicitly suggest that upregulation of JMJD6 is by copy number gain and contributes to the pathogenesis of neuroblastoma. However, authors did not demonstrate any evidence that patients with 17q gains showed upregulation of JMJD6 expression, by comparing JMJD6 expression levels between samples with and without JMJD6 copy number gain, separately within NMYC amplification positive and negative cases.

Responses:

In the original manuscript, we used the publicly available single nucleotide polymorphism (SNP) array data from 341 human neuroblastoma tissues from the Therapeutically Applicable Research to Generate Effective Treatments (TARGET) initiative of National Cancer Institute USA. Unfortunately, there is no matched gene expression data of the 341 human neuroblastoma tissues.

We have therefore switched to the human neuroblastoma tissue array-CGH (array comparative genomic hybridization) datasets from 209 patients and the matched microarray gene expression Kocak dataset which includes the 209 patients, both downloaded from the Gene Expression Omnibus website of the National Center for Biotechnology Information (SuperSeries GSE45480). Bioinformaticians in the research group of Dr. Matthias Fischer, the senior author of the datasets, have analysed the array-CGH datasets for chromosome 17q and *JMJD6* gene gain and

have analysed the Kocak microarray gene expression dataset for *JMJD6* gene expression in the 209 tumors. The analysis confirmed that *JMJD6* gene expression is higher in the human neuroblastoma tissues with chromosome 17q/*JMJD6* gene gain than those without 17q/*JMJD6* gene gain (new Figure 1c).

Among the 209 human neuroblastoma tissues, the *MYCN* oncogene is amplified in 26 tumors and non-amplified in 181 tumors. For the remaining 2 neuroblastoma tissues, *MYCN* gene amplification status is unfortunately unknown. In the 181 human neuroblastoma tissues without *MYCN* oncogene amplification, *JMJD6* gene expression was significantly higher in tumors with 17q/*JMJD6* gene gain than those without 17q/*JMJD6* gene gain (Rebuttal Figure R1a). In the 26 *MYCN*-amplified human neuroblastoma tissues, *JMJD6* gene expression was higher in tumors with 17q/*JMJD6* gene gain than those without 17q/*JMJD6* gene gain, but the difference was not statistically significant, due to the very small sample size (n=26) (Rebuttal Figure R1b). As neuroblastoma is a rare disease (9.5 cases/1,000,000 children), we could not collect enough *MYCN*-amplified human neuroblastoma tissues for chromosome copy number and *JMJD6* gene expression correlation study in several months. However, our analysis obviously demonstrates that *JMJD6* gene expression is higher in tumor tissues with chromosome 17q/*JMJD6* gene gain than those without 17q/*JMJD6* gene gain in the total population of neuroblastoma patients (new Figure 1c).

Rebuttal Figure 1

Rebuttal Figure R1. *JMJD6* is highly expressed in chromosome 17q/*JMJD6* gene-gained human neuroblastoma tissues. Chromosome 17q/*JMJD6* gene gain was identified by analysing the array-CGH datasets from 209 human neuroblastoma samples, and *JMJD6* gene expression in the 209 tumor samples was obtained from the Kocak microarray gene expression dataset. The ends of the box were the upper and lower quartiles and the whiskers were set to the minimum and maximum value within $\pm 1.5 \times$ the inter-quartile range. Two-sided unpaired Student's *t*-test was employed to analyze correlation between *JMJD6* expression and 17q/*JMJD6* gene gain status in *MYCN* gene-non-amplified (a) and *MYCN* gene-amplified tumors (b).

Changes:

- We have added the new Figure 1c to demonstrate higher levels of JMJD6 gene expression in human neuroblastoma tissues harboring chromosome 17q/*JMJD6* gene gain than human neuroblastoma tissues without 17q/*JMJD6* gene gain.
- We have revised the following sentence from line 109 to line 110 on page 5 in “**Results**” section: “.....In addition, *JMJD6* gene expression was significantly higher in 17q/*JMJD6* gene-gained than 17q/*JMJD6* gene-non-gained human neuroblastoma tissues (Fig. 1c)... ..”

Questions (2). Increased copy number of JMJD6 is found in 26% of all neuroblastoma cases and 63% of those with MYC amplification according to their SNP array data. However, over all, 17q gain is found in the majority of neuroblastoma cases (~80-90% of all neuroblastoma), where the common copy number gain involves a more telomeric region than the JMJD6 locus. In other words, 17q genes located telomeric to JMJD6 certainly showed higher frequencies of copy number gain. Authors are requested to show a map of copy number gain in 17q in their samples and compare the frequency of copy number gain of other genes.

Response:

As array-CGH data are superior to SNP array data in analysing gene copy numbers, we have switched to the human neuroblastoma tissue array-CGH datasets from 209 patients, published previously and downloaded from the Gene Expression Omnibus website of the National Center for Biotechnology Information (SuperSeries GSE45480). Bioinformaticians in the research group of Dr. Matthias Fischer, the senior author of the datasets, have analysed the array-CGH datasets for chromosome 17q and *JMJD6* gene gain. The analysis showed that the *JMJD6* gene was gained in everyone of the human neuroblastoma tissues with 17q gain, and there is no difference in the frequency of *JMJD6* gene gain and the frequency of 17q telomeric region gene gain (map of 17q gain, new Figure 1a). In addition, chromosome 17q/*JMJD6* gene was gained in 172 of the total of 209 (82.30%), including 21 of 26 *MYCN*-amplified (80.77%) and 149 of 181 *MYCN*-non-amplified (82.32%), human neuroblastoma tissues (new Figure 1b). For the two human neuroblastoma tissues with unknown *MYCN* amplification status, 17q/*JMJD6* gene was gained in both of the samples.

Changes:

- We have added chromosome 17q gain map as the new Figure 1a.
- We have added the revised data on the frequency of chromosome 17q/*JMJD6* gene gain in *MYCN*-amplified, *MYCN*-non-amplified and total human neuroblastoma tissues as the new Figure 1b.
- We have revised the following sentences from line 99 to line 109 on page 5 in “**Results**” section: “.....We analysed the human neuroblastoma array comparative genomic hybridization (array-CGH) datasets from 209 patients¹⁶⁻¹⁸ and the matched tumor tissue microarray gene expression Kocak dataset including the 209 patients¹⁸, both published previously and downloaded from the Gene Expression Omnibus website of the National Center for Biotechnology Information (SuperSeries GSE45480). Analysis of the datasets showed that the *JMJD6* gene was gained in every human neuroblastoma tissue with 17q gain, and that chromosome 17q/*JMJD6* gene was gained in 172 of the total of 209 (82.30%), including 21 of 26 *MYCN*-amplified (80.77%) and 149 of 181 *MYCN*-non-amplified (82.32%), human neuroblastoma tissues (Fig. 1a, b). For the two human neuroblastoma tissues with unknown *MYCN* amplification status, 17q/*JMJD6* gene was gained in both of the samples.....”
- We have added the following two paragraphs from line 628 to line 638 on page 27 in “**Methods**” section:

Analysis of chromosome 17q gain in human neuroblastoma tissues. Array-CGH data from 209 human neuroblastoma tissues were generated previously and downloaded from the Gene Expression Omnibus website of the National Center for Biotechnology Information (SuperSeries GSE45480)¹⁸. JMJD6 expression in the 209 human neuroblastoma tissues were also published previously as a part of the Kocak dataset and downloaded from the Gene Expression Omnibus website (SuperSeries GSE45480)¹⁸.

To create the map of chromosome 17 copy number gains, the whole chromosome was divided into bins of 100Kb and overlaps between intervals of copy number gains and these bins were counted. Only intervals exceeding a copy number of 2.5 were considered as gains. The frequency of gains among all 209 samples was plotted to show all bins across chromosome 17q.

Question (3). Authors claim a positive correlation between NMYC and JMJD6 expression levels (Fig. 1b). However, the correlation is primarily explained by higher JMJD6 expression in NMYC-amplified cases and within NMYC amplification-negative cases, almost no or very weak correlation, i.e., effect size, is observed. Similarly, despite low p-value for correlation between JMJD6 and c-Myc, the effect size is very low, if ever present.

Response:

For chromosome 17q and *JMJD6* gene copy number analysis, we have switched from a SNP array dataset to the more reliable array-CGH datasets, and have confirmed that chromosome 17q/*JMJD6* gene gain occurs at 80.77% in 26 *MYCN* gene-amplified and 82.32% in 181 *MYCN*-non-amplified human neuroblastoma tissues. The new data demonstrate that there is no difference in the frequency of 17q/*JMJD6* gene gain between *MYCN*-amplified and non-amplified human neuroblastoma tissues.

We agree with the Reviewer that the R value for the correlation between JMJD6 and N-Myc expression, as well as the R value for the correlation between JMJD6 and c-Myc expression, was low. We believe that the low R value was due to the well-known phenomena that N-Myc and c-Myc mutually suppress each other's expression, and that N-Myc and c-Myc gene expression inversely correlates with each other in human neuroblastoma tissues.

Since N-Myc and c-Myc mutually suppress each other's expression, and N-Myc and c-Myc mRNA expression is well-known to inversely correlate with each other in human neuroblastoma tissues, we used the higher value of N-Myc and c-Myc expression as the N-Myc/c-Myc expression value for each of the 476 human neuroblastoma tissues of the microarray gene expression Oberthuer dataset (two Myc as a group). We then analysed the correlation between JMJD6 mRNA expression and the higher values of N-Myc/c-Myc expression in the 476 human neuroblastoma tissues. Two-sided Pearson's correlation study revealed that JMJD6 mRNA expression positively correlated with N-Myc/c-Myc mRNA expression ($p = 8.5E-25$ and $R > 0.4$) (new Figure 1d). The data demonstrate the positive correlation between JMJD6 and N-Myc/c-Myc (two Myc as a group) expression in human neuroblastoma tissues with a good R value.

Changes:

- We have added the new data on the positive correlation between JMJD6 and N-Myc/c-Myc (two Myc as a group) mRNA expression in human neuroblastoma tissues as the new Figure 1d.
- We have added the following sentences from line 119 to line 126 on page 6 in "**Results**" section: ".....Since N-Myc and c-Myc mutually suppress each other's expression, and N-Myc and c-Myc mRNA expression is well-known to inversely correlate with each other in human neuroblastoma tissues²¹, we used the higher value of N-Myc and c-Myc expression as the N-

Myc/c-Myc expression value (two Myc as a group) for each of the 476 human neuroblastoma tissues. Two-sided Pearson's correlation study revealed that JMJD6 mRNA expression positively correlated with N-Myc/c-Myc mRNA expression with a good effect size ($R = 0.447$) in the 476 neuroblastoma tissues (Fig. 1d).....”

Question (4). If JMJD6 expression correlates with copy number, those cases showing higher JMJD6 expression are more likely to have larger deletions (gain). Thus, the apparent poor prognosis of higher JMJD6 expression might be explained by larger 17q arm segment involved in these cases, but not the slightly higher JMJD6 expression.

Response:

In the original manuscript, our Kaplan-Meier survival analysis showed that high levels of JMJD6 mRNA expression in human neuroblastoma tissues were associated with poor prognosis in the 88 and the 476 neuroblastoma patients of the Versteeg and the Oberthuer datasets respectively (Fig. 1d in the original manuscript, Fig. 1e in the revised manuscript). Furthermore, high levels of JMJD6 mRNA expression in the 405 *MYCN*-non-amplified and the 66 *MYCN*-amplified neuroblastoma tissues were also positively associated with poor patient overall survival in the large Oberthuer dataset (Fig. 1e in the original manuscript, Fig. 1f in the revised manuscript). Importantly, using the median or upper quartile of JMJD6 mRNA expression as the cut-off points, multivariable Cox regression analysis showed that high levels of JMJD6 mRNA expression was associated with poor patient overall survival and event-free survival, independent of disease stage, age at the time of diagnosis and *MYCN* amplification status (Table 1), the current key prognostic markers for neuroblastoma patients.

To address the Reviewer's question, we have performed Kaplan-Meier survival analysis of the 209 neuroblastoma patients with tumor tissue array-CGH data. When samples with segmental and numerical (whole chromosome) gain were combined, chromosome 17q gain did not associate with overall survival or event-free survival (new Supplementary Fig. 1c). When chromosome 17q segmental and numerical gains were analysed separately, 17q segmental gain was associated with poor patient overall survival ($p=0.045$), but not event-free survival ($p=0.367$), whereas 17q numerical gain was not associated with overall ($p=0.506$) or event-free ($p=0.438$) survival (new Supplementary Fig. 1d). In addition, the *MXRA7* gene is immediately upstream of the *JMJD6* gene at chromosome 17qter and is gained in the exact same manner as the *JMJD6* gene. Using the optimal cut-off level determined by Kaplan scanning, the median, upper or low quartile of *MXRA7* mRNA expression as the cut-off points, Kaplan-Meier survival analysis showed that high levels of *MXRA7* mRNA expression in human neuroblastoma tissues were not associated with poor prognosis, but were associated with better prognosis, in the 88 and the 476 neuroblastoma patients of the Versteeg and the Oberthuer datasets respectively (Supplementary Figures 2a and 2b in the revised manuscript).

Taken together, the data demonstrate that high levels of JMJD6 mRNA expression was associated with poor patient overall survival and event-free survival, due to JMJD6 expression itself and not due to larger 17q gain.

Changes:

- We have added the new data on Kaplan-Meier survival analysis of chromosome 17q segmental and/or numerical gain in neuroblastoma tissues as the new Supplementary Figures 1c and 1d.
- The data on Kaplan-Meier survival analysis of *MXRA7* mRNA expression in human neuroblastoma tissues have been re-named Supplementary Figures 2a and 2b.
- We have added/revised the following sentences from line 138 on page 6 to line 154 on page 7 on page 7 in “**Results**” section: “.....In comparison, Kaplan-Meier survival analysis of the 209

human neuroblastoma patients with array-CGH data showed that chromosome 17q gain, when samples with segmental and numerical (whole chromosome) gain were combined, did not associate with overall survival or event-free survival (Supplementary Fig. 1c). When chromosome 17q segmental and numerical gains were analysed separately, 17q segmental gain was associated with poor patient overall survival, but not event-free survival, whereas 17q numerical gain was not associated with overall or event-free survival (Supplementary Fig. 1d). In addition, the *MXRA7* gene is immediately upstream of the *JMJD6* gene at chromosome 17qter and gained in the same manner as the *JMJD6* gene (Fig. 1a). Using the optimal Kaplan scanning, the median, upper or low quartile of *MXRA7* mRNA expression as the cut-off points, Kaplan-Meier survival analysis showed that high levels of *MXRA7* mRNA expression in human neuroblastoma tissues were not associated with poor prognosis, but were associated better prognosis, in the 88 and the 476 neuroblastoma patients of the Versteeg and the Oberthuer datasets respectively (Supplementary Fig. 2a, b).

Taken together, the data suggest that the *JMJD6* gene is commonly gained in human neuroblastoma tissues, and that a high level of *JMJD6* expression independently predicts poor patient prognosis.....”

Question (5). What is the significance of the physical interaction between JMJD6 and BRD4? No data are shown supporting the role of this interaction. Also, the interaction should be supported by an opposite IP experiment, i.e., detection of BRD4 in IP with JMJD6.

Response:

In this manuscript, our microarray differential gene expression, ChIP, RT-PCR and immunoblot analyses have confirmed that JMJD6 up-regulates E2F2, N-Myc and c-Myc expression in neuroblastoma cells (revised Figures 3c and 3d, Supplementary Figures 3b-3f). In the literature, JMJD6 has been shown to activate gene transcription through forming a protein complex with BRD4 (Liu, W. et al. *Cell* 2013; 155:1581-1595) and BRD4 is well-known to activate N-Myc and c-Myc gene transcription (Zuber, J. et al. *Nature* 2011; 478:524-528 & Puissant, A. et al. *Cancer Discov* 2013; 3:308-323). To address the first part of the Review’s question, we have treated CHP134 and SK-N-AS cells with vehicle control or the BRD4 inhibitor OTX015. RT-PCR analysis confirmed that suppression of BRD4 also considerably reduced E2F2 expression (new Supplementary Figure 3g). As both components of the JMJD6-BRD4 protein complex activate E2F2 and Myc expression, our data suggest that the JMJD6-BRD4 protein complex is important for the transcription of critical oncogenes, including E2F2 and Myc.

In our original manuscript, we showed that immunoprecipitation with an anti-BRD4 antibody efficiently pulled down both BRD4 and JMJD6 proteins in CHP134 neuroblastoma cells. To address the second part of the Reviewer’s question, we have performed the opposite immunoprecipitation experiments with an anti-JMJD6 antibody in CHP134 cells. Immunoblot analysis showed that the anti-JMJD6 antibody efficiently pulled down both JMJD6 and BRD4 proteins, confirming that JMJD6 and BRD4 proteins form a protein complex in neuroblastoma cells (revised Figure 3a).

Changes:

- We have added the new immunoprecipitation data with an anti-JMJD6 antibody into Figure 3a.
- We have added the new RT-PCR data on E2F2 mRNA expression in CHP134 and SK-N-AS cells after treatment with the BRD4 inhibitor OTX015 as the new Supplementary Figure 3g.
- We have revised the following sentences from line 183 to line 187 on page 8 in “**Results**” section: “.....We performed protein co-immunoprecipitation assays with an anti-BRD4 or JMJD6 antibody or control IgG in CHP134 cells. Immunoblot analysis showed that the anti-

BRD4 and anti-JMJD6 antibodies efficiently immunoprecipitated both BRD4 and JMJD6 proteins (Fig. 3a), suggesting that JMJD6 forms a protein complex with BRD4 in neuroblastoma cells.....”

- We have added the following sentences from line 218 on page 9 to line 222 on page 10 in “**Results**” section: “.....While JMJD6 has been shown to activate gene transcription through forming a protein complex with BRD4⁷ and BRD4 is well-known to regulate N-Myc and c-Myc gene transcription^{25,26}, treatment with the small molecule BRD4 inhibitor OTX015 also considerably reduced E2F2 expression in CHP134 and SK-N-AS cells (Supplementary Fig. 3g).....”

Question (6). Reduced E2F2 binding on its target promoters in JMJD6-directed siRNA-expressed cells should be demonstrate by ChIP seq for E2F2.

Response:

We have searched the Gene Expression Omnibus website of the National Center for Biotechnology Information, and have found E2F2 ChIP sequencing dataset GSM1239497, which is a validation set for the ChIP sequencing SuperSeries GSE51142. The Abcam ab65222 anti-E2F2 antibody was used and the ChIP-seq experiment failed quality control (<https://www.ncbi.nlm.nih.gov/geo/query/acc.cgi?acc=GSM1239497>). The E2F2 ChIP sequencing data were therefore not included in the *Cell* paper (Yan J *et al.* Transcription factor binding in human cells occurs in dense clusters formed around cohesin anchor sites. *Cell*. 2013; 154:801-13). Unfortunately, no ChIP sequencing with any E2F2 antibody has ever been reported since 2013, according to the Gene Expression Omnibus website and the PubMed website.

To address the Reviewer’s question, we have treated DOX-inducible JMJD6 shRNA-2 CHP134 cells with vehicle control or DOX, followed by ChIP sequencing with a control IgG or anti-E2F2 antibody (Catalogue # sc-9967, Santa Cruz Biotechnology). Unfortunately, the anti-E2F2 antibody ChIP sequencing did not work efficiently, and bioinformatics analysis revealed only 52 peaks in vehicle control-treated samples and only 1 peak overlapped a promoter. We have also tried other anti-E2F2 antibodies for ChIP PCR. Unfortunately no anti-E2F2 antibody showed efficient binding at gene promoters. In contrast, our ChIP sequencing experiments with anti-acetylated histone H3K27 antibody, anti-RNA polymerase II antibody and anti-N-Myc antibody all worked well. We currently could not find an anti-E2F2 antibody suitable for ChIP sequencing.

Nevertheless, our Affymetrix microarray data identified E2F2 as one of the genes significantly down-regulated after JMJD6 knockdown, and gene set enrichment analysis (GSEA) of the microarray data revealed that the top transcription factor binding site consistently repressed after JMJD6 knockdown in neuroblastoma cells was the binding site for E2F (Supplementary Tables 1-4). RNA Pol II ChIP sequencing and GSEA analysis of the genes with reduced RNA Pol II binding at gene promoters after JMJD6 knockdown, showed that the top 4 gene sets were G2M checkpoint genes, E2F target genes, mitotic spindle genes and Myc target genes (Supplementary Table 6). In addition, RT-PCR and immunoblot analyses confirmed that knocking down JMJD6 with two independent siRNAs or two independent shRNAs reduced E2F2 expression, and transfection with a JMJD6 expression construct up-regulated E2F2 expression (Figure 3). Taken together, our multiple lines of evidence demonstrate that JMJD6 is critical for E2F2 expression and E2F target gene regulation.

Question (7). In Fig. 3e, N-myc is detected in control IgG lane, suspecting the precipitated N-Myc could be non-specific, but might not represent a true interaction.

Response:

To address this question, we have extracted protein from CHP134 neuroblastoma cells, and immunoprecipitated the protein with a control IgG or anti-N-Myc antibody. Immunoblot analysis confirmed that the anti-N-Myc antibody efficiently immunoprecipitated both JMJD6 and N-Myc proteins (revised Figure 3e). Together with the GST binding protein pull-down assay results in Figure 3f, the data demonstrate that JMJD6 protein directly binds to N-Myc protein.

Changes:

- We have updated Fig. 3e with the new immunoprecipitation data.

Question (8). siRNA may show negative effects on cell growth even when the results are obtained from 2 independent siRNAs. To exclude this, authors are requested to do re-expression analysis for the experiments in Fig. 4, in which JMJD6 should be overexpressed in siRNA treated cells. Authors overexpressed JMJD6 in Fig. 3d.

Response:

In the original manuscript, we showed that knocking down JMJD6 with two independent siRNAs both reduced the number of viable CHP134 and SK-N-AS cells (Alamar blue assays, Figure 4a). Please note that the two siRNAs target the 5'-untranslated region of the JMJD6 mRNA.

To address this question, we have made use of the CHP134 and SK-N-AS cells stably transfected with an empty vector or JMJD6 open reading frame (ORF) expression construct. CHP134 and SK-N-AS cells stably transfected with an empty vector or JMJD6 ORF expression construct were transfected with control siRNA, JMJD6 siRNA-1 or JMJD6 siRNA-2. Alamar blue assays showed that forced over-expression of JMJD6 largely reversed the effect of JMJD6 siRNAs in reducing the number of viable CHP134 and SK-N-AS cells (new Figure 4b). The data demonstrate that JMJD6 expression is indeed required for neuroblastoma cell proliferation/survival.

Changes:

- We have added the new Alamar blue assay data from CHP134 and SK-N-AS cells stably transfected with an empty vector or JMJD6 ORF expression construct after transfection with control siRNA, JMJD6 siRNA-1 or JMJD6 siRNA-2 as the new Figure 4b.
- We have added the following sentences from line 246 to line 251 on page 11 in “**Results**” section: “.....As the JMJD6 siRNAs targeted the 5'-untranslated region of JMJD6, CHP134 and SK-N-AS cells stably transfected with an empty vector or JMJD6 ORF expression construct were transfected with control siRNA, JMJD6 siRNA-1 or JMJD6 siRNA-2. Alamar blue assays showed that forced over-expression of JMJD6 largely reversed the effect of JMJD6 siRNAs in reducing the number of viable neuroblastoma cells (Fig. 4b).....”

Question (9). Supplementary Fig. 3b, the reduction of E2F2 expression does not correlate with the knockdown levels of N-Myc and c-Myc, raising a possibility that the reduction of E2F2 might have been off-target effect.

Response:

In the original manuscript, we showed that knocking down N-Myc expression in MYCN-amplified CHP134 cells and knocking down c-Myc in c-Myc over-expressing SK-N-AS cells reduced E2F2 mRNA (Supplementary Fig. 3a in the original manuscript, Supplementary Fig. 4a in the revised manuscript) and protein (Supplementary Fig. 3b in the original manuscript, Supplementary Fig. 4b in the revised manuscript) expression. To address the Reviewer's question, we have repeated immunoblot analysis of c-Myc and E2F2 in SK-N-AS cells after transfection with

control siRNA, c-Myc siRNA-1 or c-Myc siRNA-2 for 48 hours. Immunoblot analysis further confirmed that knocking down c-Myc with two independent siRNAs both decreased E2F2 protein expression (new Supplementary Fig. 4b). The new Supplementary Fig. 4b now shows a good correlation between N-Myc knockdown and E2F2 reduction in CHP134 cells, as well as a good correlation between c-Myc knockdown and E2F2 reduction in SK-N-AS cells.

To further address the Reviewer's question, we have treated SHEP Tet/21N neuroblastoma cells, which are stably transfected with a tetracycline/doxycycline withdrawal inducible N-Myc expression construct, with doxycycline or vehicle control. RT-PCR and immunoblot analysis confirmed that over-expression of N-Myc after doxycycline withdrawal in SHEP Tet/21N cells leads to up-regulation of E2F2 mRNA and protein expression (new Supplementary Figures 4c and 4d).

In the literature, c-Myc protein has been demonstrated to bind to Myc-responsive element E-Boxes at the E2F2 gene core promoter and directly induce E2F2 gene transcription (luciferase reporter assays with wild type and E-Box mutant E2F2 gene promoter-luciferase reporter constructs) (Sears R, Ohtani K, Nevins JR. Identification of positively and negatively acting elements regulating expression of the E2F2 gene in response to cell growth signals. *Mol. Cell. Biol.*, 1997; 17:5227-5235). We therefore did not perform ChIP assays with Myc antibodies and luciferase assays with E2F2 promoter-luciferase report constructs.

Changes:

- We have updated the immunoblot gels in the original Supplementary Fig. 3b (Supplementary Fig. 4b in the revised manuscript) and have added E2F2 RT-PCR and immunoblot data from SHEP Tet/21N cells after treatment with vehicle control or doxycycline as the new Supplementary Figures 4c and 4d.
- We have revised the following sentences from line 222 to line 227 on page 10 in "**Results**" section: ".....In addition, c-Myc is known to directly induce *E2F2* gene transcription by binding to Myc-responsive element E-Boxes at the *E2F2* gene promoter²⁷. In CHP134 and SK-N-AS cells, knocking down N-Myc or c-Myc with two independent siRNAs significantly reduced E2F2 mRNA and protein expression (Supplementary Fig. 4a, b). Conversely, up-regulation of N-Myc by DOX withdrawal in SHEP Tet/21N cells up-regulated E2F2 mRNA and protein expression (Supplementary Fig. 4c, d)....."

Question (10). THZ1 and/or panobinostat should globally affect oncogenic superenhancers, not just that of JMJD6. Why authors could conclude that the effect of these compounds on cell growth and proliferation observed *in vitro* and *in vivo* was though disruption of the enhancer at JMJD6, not others? Authors are requested to revert the suppression of cell growth and tumor progression by re-expression of JMJD6.

Response:

In the original manuscript, we showed that knocking down JMJD6 considerably reduced neuroblastoma cell proliferation, survival and clonogenic capacity *in vitro* (Figure 4) and tumour progression in mice (Figure 5), and that combination therapy with THZ1 and panobinostat synergistically reduced JMJD6 mRNA and protein expression (Figure 6d), and synergistically induced neuroblastoma cell growth inhibition and cell death (Figures 7a-d). These data suggest that THZ1 and panobinostat combination therapy is likely to exert synergistic anticancer effects through synergistically reducing JMJD6 gene expression.

To address this question, we have made use of the CHP134 and SK-N-AS cells stably transfected with an empty vector or JMJD6 ORF expression construct. The cells were treated with

vehicle control, THZ1, panobinostat, or combination of THZ1 and panobinostat. Alamar blue assays showed that over-expression of JMJD6 partially reversed the reduction in the number of viable neuroblastoma cells treated with THZ1 and panobinostat combination therapy (new Fig. 7c).

To examine whether reduction in JMJD6 expression is important for tumor growth inhibition by THZ1 and panobinostat combination therapy *in vivo*, we have xenografted two groups of 36 mice each with SK-N-AS cells stably transfected with an empty vector or JMJD6 ORF expression construct. When tumors reached 50mm³, both of the two groups of mice were divided into four sub-groups of 9 mice each and treated with vehicle control, THZ1 alone, panobinostat alone, or combination of THZ1 and panobinostat for 21 days. In the original manuscript, we treated mice with THZ1 at 10 mg/kg body weight once a day in THZ1 monotherapy group and in THZ1 plus panobinostat group. In the revised manuscript, we treated mice with THZ1 at 10 mg/kg body weight twice a day in THZ1 monotherapy group and in THZ1 plus panobinostat group. As shown in the revised Figure 7e, in mice xenografted with SK-N-AS neuroblastoma cells transfected with an empty vector, treatment with THZ1 alone or panobinostat alone reduced tumor growth, and combination therapy with THZ1 and panobinostat led to tumor regression. By contrast, in mice xenografted with SK-N-AS neuroblastoma cells transfected with a JMJD6 ORF expression construct, treatment with panobinostat reduced tumor growth, treatment with THZ1 showed no effect on tumor growth. Importantly, combination therapy with THZ1 and panobinostat was considerably less effective in mice xenografted with SK-N-AS cells transfected with a JMJD6 ORF expression construct than in mice xenografted with SK-N-AS cells transfected with an empty vector (revised Figure 7e).

Taken together, the *in vitro* and *in vivo* data confirm that reduction in JMJD6 expression is essential for the anticancer effects of THZ1 and panobinostat combination therapy against chromosome 17q/*JMJD6* gene gained neuroblastoma.

Changes:

- We have added new Figure 7c to demonstrate that transfection with a JMJD6 ORF expression construct considerably blocks the anticancer effects of THZ1 and panobinostat combination therapy against neuroblastoma cells *in vitro*.
- We have added new Figures 7e and 7f to demonstrate that ***THZ1 and panobinostat combination therapy*** synergistically reduced JMJD6 expression and ***resulted in neuroblastoma regression in mice*** xenografted with SK-N-AS cells transfected with an empty vector, and that transfection with a JMJD6 ORF expression construct considerably blocked the anticancer effect of THZ1 and panobinostat combination therapy *in vivo*.
- We have revised the following sentences from line 313 on page 13 to line 319 on page 14 in “**Results**” section: “.....We next examined whether JMJD6 was required for the anticancer effects of this combination therapy, by treating CHP134 and SK-N-AS cells stably transfected with an empty vector or JMJD6 ORF expression construct with THZ1 and panobinostat, either alone or in combination. Alamar blue assays showed that over-expression of JMJD6 partially reversed the reduction in the number of viable neuroblastoma cells after treatment with THZ1 and panobinostat combination therapy (Fig. 7c).....”
- We have added the following sentences from line 322 to line 338 on page 14 in “**Results**” section: “.....We lastly xenografted SK-N-AS neuroblastoma cells stably transfected with an empty vector or JMJD6 ORF expression construct into nude mice, and treated these mice with vehicle control, THZ1, panobinostat or combination, when tumors reached 0.05cm³. As shown in Fig. 7e, treatment with THZ1 or panobinostat alone reduced tumor growth in mice xenografted with empty vector SK-N-AS cells and treatment with panobinostat alone reduced tumor growth in mice xenografted with SK-N-AS cells transfected with JMJD6 ORF. Importantly, THZ1 and panobinostat combination therapy synergistically reduced tumor progression and resulted in tumor regression in mice xenografted with empty vector SK-N-AS

cells, but only moderately suppressed tumor progression in mice xenografted with SK-N-AS cells transfected with JMJD6 ORF (Fig. 7e). Immunoblot analysis of tumor tissues from the mice showed that THZ1 and panobinostat co-operatively reduced JMJD6 and E2F2 protein expression only in mice xenografted with empty vector SK-N-AS cells, and that transfection with the JMJD6 ORF expression construct led to considerable JMJD6 and E2F2 protein over-expression and non-response to THZ1 and panobinostat therapy (Fig. 7f). Our data therefore demonstrate that THZ1 and panobinostat combination therapy synergistically blocks neuroblastoma progression and induces tumor regression by blocking JMJD6 expression.....”

Minor comments:

Question (11). Fig. 5b: were there any censor cases? If not, Fig. 5b is a survival curve, not Kaplan-Meier estimate.

Response and *Change:*

We have changed “Kaplan-Meier survival analysis” to “survival curve analysis” for Fig. 5b in both “**Results**” and “**Figure Legends**” sections.

Question (12). Discussion is too long and mostly repeats what were described in Results. Thus, it could be substantially shortened by focusing on the significance of the results.

Response and *Change:*

We have revised “**Discussion**” section, incorporated new results and significantly shorted the section.

Question (13). JMJD6 may be important for expression of E2F2 and Myc proteins, but it is not demonstrated that it is required. So, the conclusions should be attenuated.

Response and *Change:*

We have changed “required” to “important” for modulation of E2F2 and Myc by JMJD6 in “**Abstract**”, “**Results**”, “**Discussion**” and “**Figure Legends**” sections.

REVIEWER #2:

Question 1. Authors state that there is a copy number gain of JMJD6. Are all these tumors have 17q gain? Are there any samples which have focal gain of JMJD6? It would good to show the extent of gain across the samples.

Response:

In the original manuscript, we used the publicly available single nucleotide polymorphism (SNP) array data from 341 human neuroblastoma tissues from the Therapeutically Applicable Research to Generate Effective Treatments (TARGET) initiative of National Cancer Institute USA. Unfortunately, there is no matched gene expression data from the 341 human neuroblastoma tissues. We have therefore switched to the human neuroblastoma tissue array-CGH (array comparative genomic hybridization) datasets from 209 patients with matched microarray gene

expression data (part of the Kocak dataset). The array-CGH datasets and the microarray gene expression Kocak dataset were published previously and downloaded from the Gene Expression Omnibus website of the National Center for Biotechnology Information (SuperSeries GSE45480).

Bioinformaticians in the research group of Dr. Matthias Fischer, the senior author of the datasets, have analysed the array-CGH datasets for chromosome 17q and *JMJD6* gene gain. The analysis showed that the *JMJD6* gene was gained in everyone of the human neuroblastoma tissues with 17q gain (map of 17q gain, new Figure 1a), that every neuroblastoma tissue with *JMJD6* gene gain also showed 17q gain, and that not a single tumor showed focal gain of the *JMJD6* gene. In addition, chromosome 17q/*JMJD6* gene was gained in 172 of the total of 209 (82.30%), including 21 of 26 *MYCN*-amplified (80.77%) and 149 of 181 *MYCN*-non-amplified (82.32%), human neuroblastoma tissues (new Figure 1b). For the two human neuroblastoma tissues with unknown *MYCN* amplification status, 17q/*JMJD6* gene was gained in both of the samples.

The extent of chromosome 17q/*JMJD6* gene gain across the 209 human neuroblastoma tissue samples has been plotted in the new Figure 1a.

Changes:

- We have added the chromosome 17q/*JMJD6* gene gain map across 209 human neuroblastoma tissues as the new Figure 1a.
- We have added the revised data on the frequency of chromosome 17q/*JMJD6* gene gain in *MYCN*-amplified, *MYCN*-non-amplified and total human neuroblastoma tissues as the new Figure 1b.
- We have revised the following sentences from line 99 to line 109 on page 5 in “**Results**” section: “.....We analysed the human neuroblastoma array comparative genomic hybridization (array-CGH) datasets from 209 patients¹⁶⁻¹⁸ and the matched tumor tissue microarray gene expression Kocak dataset including the 209 patients¹⁸, both published previously and downloaded from the Gene Expression Omnibus website of the National Center for Biotechnology Information (SuperSeries GSE45480). Analysis of the datasets showed that the *JMJD6* gene was gained in every human neuroblastoma tissue with 17q gain, and that chromosome 17q/*JMJD6* gene was gained in 172 of the total of 209 (82.30%), including 21 of 26 *MYCN*-amplified (80.77%) and 149 of 181 *MYCN*-non-amplified (82.32%), human neuroblastoma tissues (Fig. 1a, b). For the two human neuroblastoma tissues with unknown *MYCN* amplification status, 17q/*JMJD6* gene was gained in both of the samples.....”
- We have added the following two paragraphs from line 628 to line 638 on page 27 in “**Methods**” section:

Analysis of chromosome 17q gain in human neuroblastoma tissues. Array-CGH data from 209 human neuroblastoma tissues were generated previously and downloaded from the Gene Expression Omnibus website of the National Center for Biotechnology Information (SuperSeries GSE45480)¹⁸. *JMJD6* expression in the 209 human neuroblastoma tissues were also published previously and downloaded from the Gene Expression Omnibus website (SuperSeries GSE45480)¹⁸.

To create the map of chromosome 17 copy number gains, the whole chromosome was divided into bins of 100Kb and overlaps between intervals of copy number gains and these bins were counted. Only intervals exceeding a copy number of 2.5 were considered as gains. The frequency of gains among all 209 samples was plotted to show all bins across chromosome 17q.

Question 2. Does copy number gain correlate with RNA expression in those tumors? This is very important aspect and the data on this correlation analysis should be provided.

Response:

In the original manuscript, we used the publicly available SNP array data from 341 human neuroblastoma tissues from the Therapeutically Applicable Research to Generate Effective Treatments (TARGET) initiative of National Cancer Institute USA. Unfortunately, there is no matched gene expression data from the 341 human neuroblastoma tissues.

We have therefore switched to the human neuroblastoma tissue array-CGH datasets from 209 patients and the matched microarray gene expression Kocak dataset which includes the 209 patients, both downloaded from the Gene Expression Omnibus website of the National Center for Biotechnology Information (SuperSeries GSE45480). Bioinformaticians in the research group of Dr. Matthias Fischer, the senior author of the datasets, have analysed the array-CGH datasets for chromosome 17q and *JMJD6* gene gain and have analysed the Kocak microarray gene expression dataset for *JMJD6* gene expression. The analysis confirmed that *JMJD6* gene expression is higher in tumor tissues with chromosome 17q/*JMJD6* gene gain than those without 17q/*JMJD6* gene gain in the 209 human neuroblastoma patients (new Figure 1c).

Changes:

- We have added the new Figure 1c to demonstrate higher levels of *JMJD6* gene expression in human neuroblastoma tissues harboring chromosome 17q/*JMJD6* gene gain, compared with tumor tissues without 17q/*JMJD6* gene gain.
- We have revised the following sentence from line 109 to line 110 on page 5 in “**Results**” section: “.....In addition, *JMJD6* gene expression was significantly higher in 17q/*JMJD6* gene-gained than 17q/*JMJD6* gene-non-gained human neuroblastoma tissues (Fig. 1c)... ..”

Question 3. Authors propose a positive feedback loop between *NMYC* and *JMJD6*. However, there is a poor correlation value of 0.2 between *NMYC* and *JMJD6* in tumor datasets. Authors should explain why such poor correlation exists?

Response:

In the original manuscript, we showed that *JMJD6* expression correlates with N-Myc expression in the total cohort of neuroblastoma tissues, and correlates with c-Myc expression in *MYCN*-non-amplified neuroblastoma tissues. We agree with the Reviewer that the R value for the correlation between *JMJD6* and N-Myc expression, as well as the R value for the correlation between *JMJD6* and c-Myc expression, was low. We believe that the low R value was due to the well-known phenomena that N-Myc and c-Myc mutually suppress each other’s expression, and that N-Myc and c-Myc gene expression inversely correlates with each other in human neuroblastoma tissues.

Since N-Myc and c-Myc mutually suppress each other’s expression, and N-Myc and c-Myc mRNA expression is well-known to inversely correlate with each other in human neuroblastoma tissues, we used the higher value of N-Myc and c-Myc expression as the N-Myc/c-Myc expression value for each of the 476 human neuroblastoma tissues of the microarray gene expression Oberthuer dataset (two Myc as a group). We then analysed the correlation between *JMJD6* mRNA expression and the higher values of N-Myc/c-Myc expression in the 476 human neuroblastoma tissues. Two-sided Pearson’s correlation study revealed that *JMJD6* mRNA expression positively correlated with N-Myc/c-Myc mRNA expression ($p = 8.5E-25$ and $R > 0.4$) (new Figure 1d). The data demonstrate the positive correlation between *JMJD6* and N-Myc/c-Myc (two Myc as a group) expression in human neuroblastoma tissues with a good R value.

Changes:

- We have added the new data on the positive correlation between *JMJD6* and N-Myc/c-Myc (two Myc as a group) mRNA expression in human neuroblastoma tissues as the new Figure 1d.

- We have added the following sentences from line 119 to line 126 on page 6 in “**Results**” section: “.....Since N-Myc and c-Myc mutually suppress each other’s expression, and N-Myc and c-Myc mRNA expression is well-known to inversely correlate with each other in human neuroblastoma tissues²¹, we used the higher value of N-Myc and c-Myc expression as the N-Myc/c-Myc expression value (two Myc as a group) for each of the 476 human neuroblastoma tissues. Two-sided Pearson’s correlation study revealed that JMJD6 mRNA expression positively correlated with N-Myc/c-Myc mRNA expression with a good effect size ($R = 0.447$) in the 476 neuroblastoma tissues (Fig. 1d).....”

Question 4. Authors have used DOX inducible shRNA targeting JMJD6. However, experiments have been performed using only one shRNA. Authors should provide data from two or more shRNAs to rule out off-target effects. RNA-sequencing/microarray data should be provided from minimum of two shRNAs.

Response:

We have designed one more JMJD6 shRNA, and termed this new shRNA as JMJD6 shRNA-1 and the original shRNA as JMJD6 shRNA-2. In addition to the original doxycycline (DOX)-inducible control shRNA and JMJD6 shRNA-2 CHP134 and SK-N-AS cells, we have also established DOX-inducible JMJD6 shRNA-1 CHP134 and SK-N-AS cells.

To demonstrate that the effects of the two JMJD6 siRNAs and the original JMJD6 shRNA-2 were not off-target effects, we have treated DOX-inducible control shRNA, JMJD6 shRNA-1 and JMJD6 shRNA-2 CHP134 and SK-N-AS cells with vehicle control or DOX. RT-PCR, immunoblot, Alamar blue assays, cell cycle analysis and clonogenic assays confirmed that treatment with DOX reduced JMJD6, N-Myc, c-Myc and E2F2 mRNA and protein expression (revised Figures 3b and 3c), reduced neuroblastoma cell proliferation and clonogenic capacity, and induced apoptosis in both DOX-inducible JMJD6 shRNA-1 and JMJD6 shRNA-2, but not DOX-inducible control shRNA, CHP134 and SK-N-AS cells (revised Figures 4c and 4d, revised Supplementary Figures 5a, 5b, 5c and 5d). In addition, we have also treated DOX-inducible control shRNA and JMJD6 shRNA-1 CHP134 cells with vehicle control or DOX, followed by Affymetrix microarray experiments. Differential gene expression analysis showed that E2F2 was one of the genes significantly down-regulated by JMJD6 shRNA-1 (new Supplementary Table 1), which was consistent with the microarray data from JMJD6 shRNA-2 cells in the original manuscript (Supplementary Table 2).

Changes:

- We have added the new Affymetrix microarray gene expression data from the new DOX-inducible JMJD6 shRNA (JMJD6 shRNA-1) CHP134 cells as the new Supplementary Table 1.
- We have added the new RT-PCR and immunoblot data on reduction in E2F2, N-Myc and c-Myc mRNA and protein expression in the new DOX-inducible JMJD6 shRNA-1 CHP134 and SK-N-AS cells as the revised Figures 3b and 3c.
- We have added the new Alamar blue assay, cell cycle analysis and clonogenic assay data from the new DOX-inducible JMJD6 shRNA-1 CHP134 and SK-N-AS cells into the revised Figures 4c and 4d and Supplementary Figures 5a, 5b, 5c and 5d.
- We have revised the following sentences from line 188 on page 8 to line 195 on page 9 in “**Results**” section: “.....We next established CHP134 and SK-N-AS cells stably expressing DOX-inducible control shRNA, JMJD6 shRNA-1 or JMJD6 shRNA-2 FH1tUTG construct (Fig. 3b)²⁴. Affymetrix microarray experiments were performed in DOX-inducible control shRNA, JMJD6 shRNA-1 and JMJD6 shRNA-2 CHP134 cells 40 hours after treatment with vehicle control or DOX. Differential expression analysis identified a number of genes up- or

down-regulated after *JMJD6* gene knockdown including E2F2, which was down-regulated, following the induction of JMJD6 shRNA-1 and JMJD6 shRNA-2 (Supplementary Tables 1, 2).....”

- We have revised the following sentences from line 210 to line 214 on page 9 in “**Results**” section: “.....We next examined whether JMJD6 regulated E2F2, N-Myc and c-Myc expression. DOX-inducible control shRNA, JMJD6 shRNA-1 or JMJD6 shRNA-2 CHP134 and SK-N-AS cells were treated with vehicle control or DOX. RT-PCR and immunoblot analyses confirmed that knocking down JMJD6 significantly reduced E2F2, N-Myc and c-Myc mRNA and protein expression in CHP134 and SK-N-AS cells (Fig. 3c).....”
- We have revised the following sentences from line 251 to line 263 on page 11 in “**Results**” section: “.....Similarly, treatment with DOX resulted in a significant decrease in the number of viable DOX-inducible JMJD6 shRNA-1 and JMJD6 shRNA-2, but not DOX-inducible control shRNA, CHP134 and SK-N-AS cells (Supplementary Fig. 5a, b). Cell cycle analysis showed that treatment with DOX had no effect on the percentage of DOX-inducible control shRNA CHP134 and SK-N-AS cells at each phase of the cell cycle, but significantly increased the percentage of DOX-inducible JMJD6 shRNA-1 and JMJD6 shRNA-2 CHP134 and SK-N-AS cells at the sub-G1 phase and decreased the percentage of cells at the S phase (Fig. 4c, Supplementary Fig. 5c, d).

Colony formation assays were also performed. As shown in Fig. 4d, treatment with DOX did not have an effect on clonogenic capacity in DOX-inducible control shRNA, but considerably diminished clonogenic capacity in DOX-inducible JMJD6 shRNA-1 and JMJD6 shRNA-2, CHP134 and SK-N-AS cells. Taken together, the data suggest that JMJD6 is required for neuroblastoma cell proliferation, survival and tumorigenic capacity.....”

Question 5. Microarray analysis lacks statistical details. Authors should describe what kind filtering criteria was adapted to identify JMJD6 target genes. In fact MYCN expression is upregulated in the microarray data, which is contrary to their Western and qRT-PCR data. The decrease in MYCN at the protein level after JMJD6 KD could be due to other type of regulation; transcriptional regulation might not be a predominant mechanism. Authors need to investigate this.

Response:

We have revised the paragraph “Affymetrix microarray gene expression study” in “**Methods**” section to address this question. We have added statistical details into Supplementary Tables on microarray data.

The microarray experiments were performed only 40 hours after treatment of DOX-inducible control shRNA or JMJD6 shRNA-2 cells with vehicle control or DOX, and showed very minor increase (1.3 fold) in N-Myc expression. We have confirmed that knocking down JMJD6 with two independent siRNAs (JMJD6 siRNA-1 and JMJD6 siRNA-2) for 48 hours consistently reduced N-Myc mRNA and protein expression in CHP134 cells and reduced c-Myc mRNA and protein expression in SK-N-AS cells (revised Supplementary Figures 3e and 3f). We have also confirmed that knocking down JMJD6 with two independent DOX-inducible shRNAs (JMJD6 shRNA-1 and JMJD6 shRNA-2) for 48 hours consistently reduced N-Myc mRNA and protein expression in CHP134 cells and reduced c-Myc mRNA and protein expression in SK-N-AS cells (revised Figure 3c). Please note that the target sequences of the JMJD6 siRNAs and shRNAs are different.

To further address the Reviewer’s question, we have examined our RNA Pol II ChIP sequencing data from DOX-inducible JMJD6 shRNA-2 CHP134 cells, 40 hours after treatment with vehicle control or DOX. The ChIP sequencing data showed significantly reduced RNA Pol II

peaks at the *MYCN* gene promoter in DOX-treated samples, compared with control-treated samples (new Supplementary Figure 3b). Furthermore, we have performed ChIP PCR with an anti-RNA Pol II antibody and a control IgG in DOX-inducible JMJD6 shRNA-2 CHP134 cells after treatment with vehicle control or DOX. The ChIP PCR experiments confirmed that knocking down JMJD6 reduced RNA Pol II binding at the *MYCN* gene promoter by 66% (new Supplementary Figures 3c and 3d). The data confirm that JMJD6 up-regulates *MYCN* gene transcription and N-Myc mRNA and protein expression.

Changes:

- We have revised the paragraph “Affymetrix microarray gene expression study” on page 21 in “**Methods**” section to address this question.
- We have added statistical details into microarray data Supplementary Tables 1-4.
- We have added the new RT-PCR and immunoblot analysis of N-Myc and c-Myc mRNA and protein expression in DOX-inducible control shRNA, JMJD6 shRNA-1 and JMJD6 shRNA-2 CHP134 and SK-N-AS cells after treatment with vehicle control or DOX as the revised Figure 3c.
- We have added the new ChIP sequencing and ChIP PCR data on RNA Pol II binding at the *MYCN* gene promoter in DOX-inducible JMJD6 shRNA CHP134 cells after treatment with control or DOX as the new Supplementary Figures 3b, 3c and 3d.
- We have revised the following sentences from line 210 to line 218 on page 9 in “**Results**” section: “.....We next examined whether JMJD6 regulated E2F2, N-Myc and c-Myc expression. DOX-inducible control shRNA, JMJD6 shRNA-1 or JMJD6 shRNA-2 CHP134 and SK-N-AS cells were treated with vehicle control or DOX. RT-PCR and immunoblot analyses confirmed that knocking down JMJD6 significantly reduced E2F2, N-Myc and c-Myc mRNA and protein expression in CHP134 and SK-N-AS cells (Fig. 3c). Consistent with these data, transfection with JMJD6 siRNA-1 or siRNA-2 reduced N-Myc and c-Myc mRNA and protein expression (Supplementary Fig. 3e, f), and transfection with a JMJD6 open reading frame (ORF) expression construct led to E2F2, N-Myc and c-Myc up-regulation (Fig. 3d), in CHP134 or SK-N-AS cells.....”
- We have added the following sentences from line 205 to line 209 on page 9 in “Results” section: “.....While the *MYCN* gene was not short-listed as one of the genes with loss of RNA Pol II binding peaks at promoters in Supplementary Table 5, ChIP-Seq and ChIP PCR data revealed 60% reduction in RNA Pol II binding at the *MYCN* gene promoter after JMJD6 knockdown (Supplementary Fig. 3b, c, d).....”

Question 6. Authors have generated two stable cell lines of DOX inducible JMJD6 shRNA. However, microarray data has been provided from only one cell line. Since authors claim that JMJD6 is involved in the regulation of tumor promoting genes, such as E2F and MYCN, it would be interesting to check whether this is conserved across NB cell lines. Hence it would be good if the authors provide microarray/RNA-seq data from both CHP134 and SKNAS cell lines.

Response:

In the original manuscript, our Affymetrix microarray differential gene expression and GSEA analysis showed that the transcription factor binding site most repressed after JMJD6 knockdown in CHP134 cells was the binding site for E2F (Supplementary Table 3). To address the Reviewer’s question, we have performed triplicate Affymetrix microarray experiments in DOX-inducible control shRNA and JMJD6 shRNA-2 SK-N-AS cells, 40 hours after treatment with vehicle control or DOX. Differential gene expression and GSEA analysis revealed that the

transcription factor binding site consistently repressed after JMJD6 knockdown in SK-N-AS cells was also the binding site for E2F (Supplementary Table 4). Taken together, the microarray experiments from both CHP134 and SK-N-AS cell lines confirm that JMJD6 is important for E2F signalling pathway activity. In addition, our RT-PCR and immunoblot analysis confirmed that knocking down JMJD6 with two independent siRNAs or two independent shRNAs each significantly reduced E2F2, N-Myc and c-Myc mRNA and protein expression in CHP134 and SK-N-AS cells (revised Figure 3c and Supplementary Figures 3e and 3f), and conversely, transfection with a JMJD6 expression construct significantly increased E2F2, N-Myc and c-Myc expression in CHP134 and SK-N-AS cells (Figure 3d). Therefore, the effects of JMJD6 on E2F2 and Myc expression are not cell line-specific and conserved across cell lines.

Changes:

- We have added the new Affymetrix microarray differential gene expression and gene set enrichment analysis data from DOX-inducible control shRNA and JMJD6 shRNA-2 SK-N-AS cells as the new Supplementary Table 4.
- We have added the following sentences from line 195 to line 199 on page 9 in **Results** section: “.....We also performed Affymetrix microarray experiments in DOX-inducible control shRNA and JMJD6 shRNA-2 SK-N-AS cells 40 hours after treatment with vehicle control or DOX. Differential gene expression and gene set enrichment analysis (GSEA) revealed that the transcription factor binding site consistently repressed after JMJD6 knockdown in both CHP134 and SK-N-AS cell lines was the binding site for E2F (Supplementary Tables 3, 4).....”

Question 7. RNAP II ChIP-sequencing data lacks details and statistical information. XL sheets (Tables) lack information such as peak location, peak score, fold change and also the information of peaks existence in individual replicates. In particular I found the RNAP II peak consideration between control and DOX samples a bit strange (Genes with reduced RNA Pol II binding were therefore selected as those that had a RNA Pol II ChIP-seq peak called within the promoter in both replicates in vehicle control treated samples, but no peak in at least one replicate in DOX-treated samples). I am not sure whether it is correct way to consider the peaks differentially for control and DOX samples. Another important aspect is that is there any overlap between genes downregulated upon JMJD6 KD with loss of RNAP II peaks following JMJD6 loss? The presented data do not provide any insight how JMJD6 regulates gene transcription by MYCN interaction.

Response:

We have added information on peak location, average peak score (normalised reads/kilobase), average fold change and statistics to the RNA Pol II ChIP sequencing data table (revised Supplementary Table 5) from DOX-inducible JMJD6 shRNA CHP134 cells. RNA Pol II binding peaks were clearly identified at the promoters of all genes listed in Supplementary Table 5 in all control-treated samples, but not in anyone of the DOX-treated samples. We have revised the “ChIP sequencing” paragraph in “**Methods**” section to provide more details about ChIP sequencing data analysis.

We have analysed overlap between the 789 genes down-regulated by ≥ 1.5 fold upon JMJD6 knockdown (Supplementary Table 2) and 664 genes with considerable reduction in RNA Pol II peaks following JMJD6 knockdown (Supplementary Table 5) in DOX-inducible JMJD6 shRNA-2 CHP134 cells. Forty genes were found to show both reduced gene expression and loss of RNA Pol II binding at gene promoters. The low rate of overlap of the two gene sets can be

explained by the phenomena that: (1) reduction in gene/mRNA expression takes place after reduction in RNA Pol II binding; (2) reduction in mRNA expression, as identified by Affymetrix microarray, can be due to reduced gene transcription, reduced mRNA stability or increased mRNA degradation.

As shown in Supplementary Table 6, E2F and Myc target gene sets were among the top 4 JMJD6 target gene sets, according to RNA Pol II ChIP sequencing and GSEA analysis of genes with reduced RNA Pol II binding at gene promoters, after JMJD6 knockdown. Our protein co-immunoprecipitation assays and GST binding protein pull-down assays confirmed that JMJD6 protein directly binds to N-Myc protein at the Myc Box II region (Figures 3e and 3f). In the literature, N-Myc regulates gene transcription mainly through binding to target gene promoters, and JMJD6 activates gene transcription by interacting with MED12 in the mediator complex and interacting with BRD4 at transcriptional super-enhancers. Our data therefore suggest that JMJD6 activates gene transcription through binding to BRD4 and N-Myc, and that the interaction between JMJD6 and N-Myc is likely to link transcriptional enhancers and promoters for more efficient transcriptional activation.

Changes:

- We have added information on peak location, average peak score (normalized reads/kilobase), average fold change and statistics to the RNA Pol II ChIP sequencing data table (revised Supplementary Table 5).
- We have revised the table on GSEA analysis of genes with reduced RNA Pol II binding at gene promoters after JMJD6 knockdown (revised Supplementary Table 6).
- We have added the following sentences from line 200 to line 205 on page 9 in “**Results**” section: “.....To examine whether JMJD6 directly regulates gene transcription, we performed ChIP sequencing (ChIP-Seq) experiments in DOX-inducible JMJD6 shRNA-2 CHP134 cells. The ChIP-Seq data identified a list of genes with considerably reduced RNA polymerase II (RNA Pol II) binding peaks at their gene promoters (Supplementary Table 5). GSEA analysis of the repressed genes showed that the top 4 gene sets were G2M checkpoint genes, E2F target genes, mitotic spindle genes and Myc target genes (Supplementary Table 6).....”
- We have added the following sentences from line 367 to line 373 on page 16 in “**Discussion**” section: “.....In the literature, N-Myc regulates gene transcription mainly through binding to target gene promoters, and JMJD6 activates gene transcription by interacting with MED12 in the mediator complex and interacting with BRD4 at transcriptional super-enhancers^{7,31}. Our data therefore suggest that JMJD6 activates gene transcription partly through binding to N-Myc, and that the interaction between JMJD6 and N-Myc is likely to link transcriptional enhancers and promoters for more efficient gene transcription.....”

Question 8. Since authors data claims that JMJD6 and NMYC functionally associated to regulate global gene expression, it would be interesting to see their colocalization across the genome (ChIP-seq) and also the functional overlap among the target genes of JMJD6 and NMYC to reinforce their common effects on global gene regulation in neuroblastoma. Does their colocalization specifically occurs at the super-enhancers associated with oncogenic drivers?

Responses

In the literature, anti-JMJD6 antibody from Abcam (catalogue number: ab10526) has been successfully used for ChIP sequencing experiments (Liu W et al. *Cell* 2013; 155:1581-1595 & Gao WW et al. *Mol Cell* 2018; 70:340-357). Unfortunately, Abcam has discontinued the antibody, and no anti-JMJD6 antibody targeting the same peptides is available.

To address this question, we have performed ChIP sequencing with control IgG, anti-acetylated histone H3K27, anti-BRD4 antibody, anti-N-Myc antibody, anti-JMJD6 antibody number 1 (Santa Cruz Biotechnology, H-7, catalogue number sc-28348) and anti-JMJD6 antibody number 2 (Invitrogen, catalogue number PA1-24739) in CHP134 cells. Unfortunately, ChIP sequencing with the two anti-JMJD6 antibodies did not generate reliable peaks.

We then analysed acetylated H3K27 antibody, BRD4 antibody and N-Myc antibody ChIP sequencing data. Consistent with the literature, the anti-N-Myc antibody mainly enriched DNA fragments at gene promoter regions (1456 genes with peaks at their gene promoters), and the anti-BRD4 antibody mainly enriched DNA fragments at super-enhancer regions in CHP134 neuroblastoma cells. To further understand the potential role of N-Myc in super-enhancers, compared with normal enhancers, we compared normalised ChIP-sequencing reads per kilobase DNA for BRD4 and N-Myc ChIP sequencing data. We found that the BRD4 antibody considerably enriched more DNA fragments at transcriptional super-enhancers, while the N-Myc antibody significantly less enriched DNA fragments at super-enhancers, than normal enhancers ($p < 0.0001$) (Rebuttal Figure R2). The data suggest that N-Myc does not play an important role in super-enhancer activity in CHP134 neuroblastoma cells.

Rebuttal Figure R2. N-Myc does not preferentially binds to transcriptional super-enhancers in CHP134 neuroblastoma cells. Enhancers ($n = 13,595$) and super-enhancers ($n = 182$) were defined with H3K27ac ChIP-seq data using MACS (PMID: 18798982) and ROSE (PMID: 23582323), as described (PMID: 30371817). The number of ChIP-seq reads covering each region was determined using BEDTools (PMID: 25199790) and the number of reads from each ChIP-seq experiment was determined using SAMtools (PMID: 19505943). “Normalised ChIP-seq reads per kilobase” were determined using the following formula: *Total number of reads overlapping region of interest / size of region of interest (base pairs) / normalisation factor * 1000*, where *normalisation factor = number of ChIP-seq reads in file / average number of ChIP-seq reads across all files*. Significance was determined using a two-sided Mann Whitney U-Test.

In the literature, JMJD6 protein has been shown to activate gene transcription through binding to BRD4 at super-enhancers, not promoters (Liu W et al. *Cell* 2013; 155:1581-1595). We have confirmed that JMJD6 directly binds to N-Myc protein at Myc Box II (Figures 3e and 3f), and that Myc target gene set is one of the top four gene sets with considerably reduced RNA Pol II binding at gene promoters, after JMJD6 knockdown (Supplementary Table 6). We therefore suggest that JMJD6 activates gene transcription through binding to BRD4 and N-Myc, and that the interaction between JMJD6 and N-Myc is likely to link transcriptional enhancers and promoters for more efficient transcriptional activation.

Changes:

- We have added the following sentences from line 367 to line 373 on page 16 in “**Discussion**” section: “.....In the literature, N-Myc regulates gene transcription mainly through binding to target gene promoters, and JMJD6 activates gene transcription by interacting with MED12 in the mediator complex and interacting with BRD4 at transcriptional super-enhancers^{7,31}. Our data therefore suggest that JMJD6 activates gene transcription partly through binding to N-Myc, and that the interaction between JMJD6 and N-Myc is likely to link transcriptional enhancers and promoters for more efficient gene transcription.....”

Question 9. Figure 7f: I do not see any noticeable decrease of JMJD6, CMYC and E2F2 upon THZ1 in SKNAS xenografts while figure 6 does show effect on these proteins. Why this discrepancy? What is the IC50 concentration of THZ1 for CHP134 and SKNAS cell lines? E2F2 is shown as single band in many immunoblots, is shown as two bands in THZ1 treatment in Figure 7f. This is also observed with JMJD6 and E2F2. There is an inconsistency in the presentation of the blots.

Response:

We agree with the reviewer that, in the original Figure 7f, THZ1 treatment alone did not significantly reduce JMJD6 and E2F2 protein expression in the SK-N-AS xenograft tumor tissues. This is likely due to the short half-life of THZ1 in mice and the mice were treated with THZ1 only once per day. The IC50 concentration of THZ1 for CHP134 and SKNAS cell lines were 55.61 nM and 30.74 nM respectively.

To address this question, we have xenografted two groups of 36 mice each with SK-N-AS cells stably transfected with an empty vector or JMJD6 ORF expression construct. When tumors reached 50mm³, both of the two groups of mice were divided into four sub-groups of 9 mice each and treated with vehicle control, THZ1 alone, panobinostat alone, or combination of THZ1 and panobinostat for 21 days. In the original manuscript, we treated mice with THZ1 at 10 mg/kg body weight *once a day* in THZ1 monotherapy group and in THZ1 plus panobinostat group. In the revised manuscript, we treated mice with THZ1 at 10 mg/kg body weight *twice a day* in THZ1 monotherapy group and in THZ1 plus panobinostat group. As shown in the new Figures 7e, in mice xenografted with empty vector SK-N-AS neuroblastoma cells, treatment with THZ1 alone or panobinostat alone reduced tumor growth, and **combination therapy with THZ1 and panobinostat led to tumor regression**. By contrast, in mice xenografted with SK-N-AS neuroblastoma cells transfected with a JMJD6 ORF expression construct, treatment with panobinostat reduced tumor growth, treatment with THZ1 showed no effect on tumor growth. Importantly, combination therapy with THZ1 and panobinostat was considerably less effective in mice xenografted with SK-N-AS cells transfected with a JMJD6 ORF expression construct than in mice xenografted with SK-N-AS cells transfected with an empty vector (revised Figure 7e). Immunoblot analysis showed that treatment with THZ1 reduced JMJD6 and E2F2 protein expression, and treatment with THZ1 plus panobinostat synergistically reduced JMJD6 and E2F2 protein expression, in empty vector SK-N-

AS xenograft tumor tissues. By contrast, immunoblot analysis showed that tumor tissues from mice xenografted with SK-N-AS cells transfected with a JMJD6 ORF construct exhibit considerably higher levels of JMJD6 and E2F2 protein expression, and treatment with THZ1 or combination of THZ1 and panobinostat did not have a significant effect.

The Reviewer pointed out that E2F2 and JMJD6 proteins were shown as single bands in cell culture experiments, but double bands in mouse tumor tissue immunoblot. As JMJD6 protein band was very close to E2F2 protein band, in the revised manuscript, we loaded protein to separate gels, and performed JMJD6 and E2F2 immunoblot with the different membranes, so as to avoid non-specific bands due to incomplete stripping of previous antibodies and immunoblot bands. As shown in the revised Figure 7f, there is only one band for JMJD6 and one band for E2F2 in the immunoblot data.

Taken together, the *in vitro* and *in vivo* data confirm that reduction in JMJD6 expression is essential for the dramatic anticancer effects of THZ1 and panobinostat combination therapy against chromosome 17q/*JMJD6* gained neuroblastoma.

Changes:

- We have added the following sentences from line 322 to line 338 on page 14 in “**Results**” section: “.....We lastly xenografted SK-N-AS neuroblastoma cells stably transfected with an empty vector or JMJD6 ORF expression construct into nude mice, and treated these mice with vehicle control, THZ1, panobinostat or combination, when tumors reached 0.05cm³. As shown in Fig. 7e, treatment with THZ1 or panobinostat alone reduced tumor growth in mice xenografted with empty vector SK-N-AS cells and treatment with panobinostat alone reduced tumor growth in mice xenografted with SK-N-AS cells transfected with JMJD6 ORF. Importantly, THZ1 and panobinostat combination therapy synergistically reduced tumor progression and resulted in tumor regression in mice xenografted with empty vector SK-N-AS cells, but only moderately suppressed tumor progression in mice xenografted with SK-N-AS cells transfected with JMJD6 ORF (Fig. 7e). Immunoblot analysis of tumor tissues from the mice showed that THZ1 and panobinostat co-operatively reduced JMJD6 and E2F2 protein expression only in mice xenografted with empty vector SK-N-AS cells, and that transfection with the JMJD6 ORF expression construct led to considerable JMJD6 and E2F2 protein over-expression and non-response to THZ1 and panobinostat therapy (Fig. 7f). Our data therefore demonstrate that THZ1 and panobinostat combination therapy synergistically blocks neuroblastoma progression and induces tumor regression by blocking JMJD6 expression.....”

Question 10. The use of CDK7 inhibitor in the current investigation is not clear. Moreover, the use of CDK7 inhibition for the MYCN driven tumor is not novel (<https://www.ncbi.nlm.nih.gov/pubmed/2541650>). CDK7 is bound to a subset of the super enhancers (<https://www.nature.com/articles/nature13393>) and regulate the expression of oncogenes such as RUNX1. Is the super enhancer that regulates JMJD6 bound by Cdk7?

Response:

Why using CDK7 inhibitor? We have confirmed that the *JMJD6* gene in chromosome 17q-gained neuroblastoma cells is associated with transcriptional super-enhancers (Figures 6a and 6b). In the literature, CDK7 selectively up-regulates the expression of super-enhancer-associated oncogenes, and CDK7 inhibitors selectively repress the expression of super-enhancer-associated oncogenes (Kwiatkowski, N. et al. Targeting transcription regulation in cancer with a covalent CDK7 inhibitor. *Nature* 2014; 511:616-620 & Chipumuro E et al. CDK7 Inhibition Suppresses

Super-Enhancer-Linked Oncogenic Transcription in MYCN-Driven Cancer. *Cell* 2014; 159:1126-1139). We therefore used the CDK7 inhibitor THZ1 to suppress JMJD6 gene expression (as shown in Figures 6c and 6d).

Novelty of this study. While CDK7 inhibition of MYCN-driven neuroblastoma has been reported, in this manuscript, we have identified combination therapy with the CDK7 inhibitor THZ1 and the HDAC inhibitor panobinostat as a much more efficacious strategy to synergistically block JMJD6 gene expression and consequently block E2F2, N-Myc and c-Myc expression (Figure 6d and Supplementary Figures 7a and 7b). Importantly, we have confirmed that ***THZ1 and panobinostat combination therapy leads to tumour regression in neuroblastoma-bearing mice***, and that the anticancer effect of the combination therapy is considerably reversed in mice when the neuroblastoma cells were transfected with a JMJD6 ORF expression construct (Figure 7e), which does not respond to THZ1 treatment.

CDK7 binding to JMJD6 gene super-enhancers. We have performed ChIP assays with a control IgG or anti-CDK7 antibody, followed by PCR with primers targeting the JMJD6 gene super-enhancer or promoter in CHP134 neuroblastoma cells. The ChIP assays showed that CDK7 considerably more abundantly bound to the super-enhancer region than the promoter region (new Supplementary Figure 6a).

Changes:

- We have added the control IgG and anti-CDK7 antibody ChIP PCR data as the new Supplementary Figure 6a.
- We have added the following sentence from line 289 to line 290 on page 12 in “**Results**” section: “.....ChIP PCR showed that CDK7 bound more abundantly to the *MXRA7/JMJD6* gene super-enhancer region than the *JMJD6* gene promoter region (Supplementary Fig. 6a).....”

MINOR COMMENTS:

Minor Point 1. In Figure 2A, last panel C-MYC KD in SKNAS cells does not look biological although significance is provided.

Responses. Figure 2a demonstrated that knocking down N-Myc expression in CHP134 cells, and knocking down c-Myc expression in SK-N-AS cells, with two independent N-Myc siRNAs and two independent c-Myc siRNAs reduced JMJD6 mRNA expression. Consistent with these data, N-Myc siRNAs and c-Myc siRNAs significantly reduced JMJD6 protein expression in CHP134 and SK-N-AS cells (Figure 2b), and forced N-Myc over-expression in SHEP tet/21N cells resulted in JMJD6 mRNA and protein up-regulation (Figure 2c). Moreover, ChIP assays demonstrated that N-Myc and c-Myc proteins bound to the JMJD6 gene promoter (Figures 2d and 2e). Taken together, the data confirm that N-Myc and c-Myc up-regulate JMJD6 mRNA and protein expression.

Minor Point 2. Figure 2B: Authors should quantify Western bands.

Responses

We have quantified immunoblot gels from triplicate experiments for Figure 2b. Protein quantification confirmed that transfection with N-Myc siRNAs in CHP134 cells and transfection with c-Myc siRNAs in SK-N-AS cells significantly reduced JMJD6 protein expression (new Supplementary Figure 3a).

Changes

We have added the Figure 2b protein quantification data as the new Supplementary Figure 3a.

Minor Point 3. In Fig 6A: authors highlight enhancer-specific modification peaks in *MXRA7* gene but not *JMJD6*? Does these hypothetical enhancers are specific to *JMJD6* or *MXRA7* or both?

Responses

We have treated CHP134 and SK-N-AS cells with vehicle control or 32 or 64 nM THZ1, which specifically suppresses the expression of super-enhancer-controlled genes, for 24 hours. RT-PCR analysis showed that treatment with the CDK7/super-enhancer inhibitor THZ1 reduced *JMJD6* gene expression (Figure 6c), but showed no effect on *MXRA7* gene expression in CHP134 and SK-N-AS cells (new Supplementary Figure 6b). The data demonstrate that the super-enhancers at the *MXRA7/JMJD6* gene region in CHP134 and SK-N-AS cells up-regulates *JMJD6* but not *MXRA7* gene expression.

Changes:

- We have added the new RT-PCR data on *MXRA7* mRNA expression in CHP134 and SK-N-AS cells after treatment with vehicle control or THZ1 as the new Supplementary Figure 6b.
- We have added the following sentences from line 291 on page 12 to line 296 on page 13 in “**Results**” section: “.....We then investigated the effect of the CDK7/super-enhancer inhibitor THZ1 on *JMJD6* and *MXRA7* gene expression. RT-PCR and immunoblot analyses showed that treatment with THZ1 reduced *JMJD6* and N-Myc expression in CHP134 cells, and *JMJD6* and c-Myc expression in SK-N-AS cells (Fig. 6c), but showed no effect on *MXRA7* expression (Supplementary Fig. 6b). The data suggest that the *MXRA7/JMJD6* gene super-enhancers activate the transcription of the *JMJD6* but not *MXRA7* gene.”

Minor Point 4. Check the following sentence “demethylating histone H4 arginine 3 (H4R3)”.

Response and Changes:

We have revised “demethylating histone H4 arginine 3” to “demethylating histone H4 at arginine 3” in the manuscript.

Tao Liu

Reviewers' comments:

Reviewer #1 (Remarks to the Author):

In their revised manuscript, authors address most of the issues this reviewer raised. I have a few remaining issues, which should be further addressed.

1) Authors demonstrated a significant difference in JMJD6 expression between tumors with and without 17q gains. However, the effect size of 17q gain is very small if ever, which raised a serious concern that that unlike authors' claim, the effect of 17q gain on JMJD6 expression is very small. In fact, the effect on patient survival was only found for increased JMJD6 expression but not 17q gains, even though all 17q gains affected JMJD6. Thus, it seems to be difficult to ascribe elevated JMJD6 to 17q gain.

2) Meanwhile, 17q gain is by far the most frequent genetic abnormality in neuroblastoma and characterize this unique pediatric tumor, suggesting its important role in the pathogenesis of neuroblastoma. The common region of 17q involves a large 17q segment, which contains many genes, suggesting trisomy of multiple genes are important. Authors showed that expression of MXRA7, another gene on 17q, did not correlate with survival. Of course, there is a fine line between the fact that a gene is involved in neuroblastoma pathogenesis and that it affects survival.

To summarize, considering the issues 1) and 2), it is misleading to state as if JMJD6 is the gene target of 17q gain that is important for neuroblastoma pathogenesis and prognosis.

Reviewer #2 (Remarks to the Author):

The revised manuscript is much improved. In the revised version, authors have convincingly addressed functional connection between JMJD6 copy number alterations and JMJD6 expression. However, there are several concerns remain unaddressed.

Authors provide microarray data from shRNA-1 and shRNA-2. FDR values seems to be quite for many genes and authors have not filtered genes based on p value and FDR. Moreover, authors should integrate shRNA-1 and shRNA-2 data and perform GSEA and other functional enrichment analyses

with genes showing significant differential expression with both shRNAs (Fold change, p value and FDR) genes. Surprisingly, E2F2 does not show p-value and FDR (FC: -1.83 P value 0.19015 FDR 0.9981599 E2F2) in shRNA-1 KD and show only shRNA-2 KD. Furthermore, shRNA-1 KD sample, JMJD6 downregulation does not show any significance (-1.005 -2.01 P value 0.1061 FDR: 0.9981599).

I cannot find MYCN as a differentially expressed RNA in the microarray data. Authors present experimental evidence (qRT-PCR and Western) in the revised Figure 3C that MYCN is a transcriptional target of JMJD6.

In GSEA analysis, authors show that enrichment of NMYC-01 gene set upon JMJD6 knockdown in supplementary table 3. However, the enrichment is not significant (FDR value: 0.48).

Overall, microarray data in the revised version is not well controlled. Information about the replicates for each microarray experiment is lacking. FDR cut off at 0.25 is too high to obtain reliable data.

The authors have not provided crucial data on JMJD6/MYCN (ChIP-seq) binding sites across the genome. Authors have also not addressed genome-wide E2F2 binding sites in control and JMJD6 shRNA KD cells, as reviewer 1 suggested.

Reviewer #1:

1) Authors demonstrated a significant difference in JMJD6 expression between tumors with and without 17q gains. However, the effect size of 17q gain is very small if ever, which raised a serious concern that that unlike authors' claim, the effect of 17q gain on JMJD6 expression is very small. In fact, the effect on patient survival was only found for increased JMJD6 expression but not 17q gains, even though all 17q gains affected JMJD6. Thus, it seems to be difficult to ascribe elevated JMJD6 to 17q gain.

Response:

In the first revision manuscript, we combined tumors with chromosome 17q segmental gain and tumors with 17q numerical (whole chromosome) gain as the 17q gain group, and found that *JMJD6* gene expression was slightly higher in 17q-gained than 17q-non-gained human neuroblastoma tissues (original Fig. 1c), but the effect size was small.

To address the reviewer's question, we separated tumors with chromosome 17q segmental gain from tumors with 17q numerical gain, and examined whether 17q segmental gain and numerical (whole chromosome) gain had different effects on 17q copy number and *JMJD6* gene expression. While tumors with 17q segmental gain or numerical gain showed higher 17q copy

number than tumors without 17q gain, tumors with 17q segmental gain also showed significantly higher 17q copy number than tumors with 17q numerical gain (new Fig. 1c). Consistent with these data, *JMJD6* gene expression was significantly higher in tumors with 17q segmental gain than in tumors with 17q numerical gain and tumors without 17q gain, and there was no difference in *JMJD6* gene expression between tumors with 17q numerical gain and tumors without 17q gain (new Fig. 1d). Please note that the effect size of 17q segmental gain on increased 17q copy number and increased *JMJD6* expression is good (new Fig. 1c and new Fig. 1d). *The data suggest that 17q segmental gain leads to significantly increased 17q copy number and significantly higher JMJD6 gene expression, and that 17q numerical gain leads to slightly increased 17q copy number which is not enough to increase JMJD6 gene expression.*

In the first revision manuscript, tumors with chromosome 17q segmental or numerical gain were combined as the 17q gain group for Kaplan-Meier survival analysis of 209 human neuroblastoma patients. The analysis showed that 17q gain, compared with 17q non-gain, did not associate with patient overall survival (Supplementary Fig. 1c). Importantly, when we analyzed tumors with chromosome 17q segmental gain (111 cases) and tumors with 17q numerical gain (61 cases) separately against tumors without 17q gain (37 cases), Kaplan-Meier survival analysis showed that 17q numerical gain was not associated with patient overall survival ($p=0.506$), but 17q segmental gain was significantly associated with poor patient overall survival ($p=0.045$) (Supplementary Fig. 1d).

Taken together, the new data demonstrate that 17q segmental gain, but not numerical gain, in human neuroblastoma tissues significantly increases 17q copy number and *JMJD6* gene expression and correlates with poor patient prognosis. This is consistent with our finding that high levels of *JMJD6* gene expression in human neuroblastoma tissues also correlate with poor patient prognosis (Fig. 1f, g).

Changes:

- We have added new data on the different effects of chromosome 17q segmental gain and 17q numerical gain on 17q copy number and *JMJD6* gene expression as the new Figures 1c and 1d. Kaplan-Meier survival analysis of the prognostic value of 17q segmental gain, 17q numerical gain and 17q non-gain was listed as Supplementary Figure 1d.
- We have added the following sentences as the second paragraph on page 5 in “**Results**” section: “.....We next examined whether chromosome 17q segmental gain and numerical (whole chromosome) gain had different effects on 17q copy number and *JMJD6* gene expression. While tumors with 17q segmental gain or numerical gain showed higher 17q copy number than tumors without 17q gain, tumors with 17q segmental gain also showed significantly higher 17q copy number than tumors with 17q numerical gain (Fig. 1c). Consistent with these data, *JMJD6* gene expression was significantly higher in tumors with 17q segmental gain than in tumors with 17q numerical gain or without 17q gain, and there was no difference in *JMJD6* gene expression between tumors with 17q numerical gain and tumors without 17q gain (Fig. 1d).....”
- We have revised the first paragraph on page 7 in “**Results**” section to the following: “.....In comparison, Kaplan-Meier survival analysis of the 209 human neuroblastoma patients with array-CGH data showed that chromosome 17q gain, when samples with segmental or numerical gain were combined, did not associate with patient overall survival (Supplementary Fig. 1c). When tumors with chromosome 17q segmental gain or numerical gain were analyzed separately, 17q segmental gain was associated with poor patient overall survival, whereas 17q numerical gain was not associated with patient overall survival (Supplementary Fig. 1d).....”
- We have revised the following sentences from the 7th line 7 to the 11th line on page 16 in “**Discussion**” section: “.....and that 17q segmental gain leads to higher 17q copy number and higher *JMJD6* gene expression than 17q numerical gain and 17q no gain. Importantly,

chromosome 17q segmental gain and high levels of JMJD6, but not 17q numerical gain or high levels of *MXRA7*, the gene immediately upstream of *JMJD6* at chromosome 17qter, in neuroblastoma tissues correlates with poor patient prognosis.....”

2) Meanwhile, 17q gain is by far the most frequent genetic abnormality in neuroblastoma and characterizes this unique pediatric tumor, suggesting its important role in the pathogenesis of neuroblastoma. The common region of 17q involves a large 17q segment, which contains many genes, suggesting trisomy of multiple genes are important. Authors showed that expression of *MXRA7*, another gene on 17q, did not correlate with survival. Of course, there is a fine line between the fact that a gene is involved in neuroblastoma pathogenesis and that it affects survival.

To summarize, considering the issues 1) and 2), it is misleading to state as if *JMJD6* is the gene target of 17q gain that is important for neuroblastoma pathogenesis and prognosis.

Response:

We agree with the reviewer that chromosome 17q gain is the most frequent genetic abnormality and plays its important role in the pathogenesis of neuroblastoma. We have found that *JMJD6* is highly expressed in human neuroblastoma tissues with 17q segmental gain (new Fig. 1d), and that both high levels of *JMJD6* gene expression (Fig. 1f and Fig. 1g) and 17q segmental gain (Supplementary Fig. 1d) are associated with poor patient prognosis. We have also found that high levels of *MXRA7*, the gene immediately up-stream of the *JMJD6* gene, in human neuroblastoma tissues does not correlate with poor patient survival (Supplementary Fig. 2).

We have demonstrated that knocking down *JMJD6* with two independent siRNAs or two independent shRNAs considerably reduces chromosome 17q/*JMJD6* gene-gained neuroblastoma cell proliferation (Fig. 4a-4c) and clonogenic capacity (Fig. 4d), and induces apoptosis (Fig. 4c). Conversely, forced *JMJD6* high expression significantly increases neuroblastoma cell proliferation (Fig. 4b). Importantly, in mice xenografted with doxycycline-inducible *JMJD6* shRNA neuroblastoma cells harboring 17q/*JMJD6* gene gain, knocking down *JMJD6* gene expression with doxycycline in food considerably suppresses tumor progression and improves mouse survival (Figure 5).

Considering the above multiple lines of evidence, we would like to conclude that *JMJD6* gene expression is increased due to chromosome 17q segmental gain, and that high levels of *JMJD6* gene expression is a contributing factor for neuroblastoma pathogenesis and poor prognosis. Nevertheless, we agree with the reviewer that other genes at 17q also play important roles, and we have therefore made the below revision of the manuscript.

Changes:

- We have revised the following sentences from the 12th line to the 17th line in the second paragraph on page 17 in “**Discussion**” section to the following: “.....we have confirmed that knocking down *JMJD6* considerably reduces neuroblastoma cell proliferation, induces apoptosis, and dramatically reduces clonogenic capacity. Consistent with these *in vitro* data, knocking down *JMJD6* reduces E2F2 and Myc expression, suppresses tumor progression and improves survival in mice. The data suggest that high levels of *JMJD6* expression due to chromosome 17q21-ter segmental gain contributes to neuroblastoma pathogenesis.. ..”

REVIEWER #2:

Question 1. Authors provide microarray data from shRNA-1 and shRNA-2. FDR values seems to be quite for many genes and authors have not filtered genes based on p value and FDR. Moreover, authors should integrate shRNA-1 and shRNA-2 data and perform GSEA and other functional enrichment analyses with genes showing significant differential expression with both shRNAs (Fold change, p value and FDR) genes. Surprisingly, E2F2 does not show p-value and FDR (FC: -1.83 P value 0.19015 FDR 0.9981599 E2F2) in shRNA-1 KD and show only shRNA-2 KD. Furthermore, shRNA-1 KD sample, JMJD6 downregulation does not show any significance (-1.005 -2.01 P value 0.1061 FDR: 0.9981599).

Response:

To address this question and to achieve better p values and FDR values, we have repeated the microarray experiments so that we now have 4 replicates of microarray data from doxycycline (DOX)-inducible control shRNA, JMJD6 shRNA-1 and JMJD6 shRNA-2 CHP134 cells. We have analyzed the 4 replicates of microarray data by integrating data from DOX-inducible JMJD6 shRNA-1 and shRNA-2 cells, filtered genes with fold changes of > 1.5, p values < 0.05 and FDR values < 0.20 by both JMJD6 shRNA-1 and JMJD6 shRNA-2 (new Supplementary Table S1), and perform GSEA analyses (new Supplementary Table S2). The data showed that E2F2 expression was down-regulated by the two JMJD6 shRNAs by 1.84 fold with p = 0.00013 and FRD = 0.011 (fold change = -1.57, p = 0.0017, FDR = 0.172 for JMJD6 shRNA-1; and FC = -2.46; p = 3.37E-05, FDR = 0.0068 for JMJD6 shRNA-2). JMJD6 expression was down-regulated by the two JMJD6 shRNAs by 1.81 fold with p = 2.89E-06 and FRD = 0.00098 (fold change = -1.63, p = 0.00047, FDR = 0.106 for JMJD6 shRNA-1; and FC = -2.05; p = 8.19E-06, FDR = 0.0033 for JMJD6 shRNA-2). Consistent with the 1st revision manuscript, GSEA analysis showed that the transcription factor binding site consistently repressed after JMJD6 knockdown with shRNA-1 or shRNA-2 was the binding site for E2F (new Supplementary Table 2).

Changes:

- We have integrated the 4 replicates of new microarray data from DOX-inducible control shRNA-1, JMJD6 shRNA-1 and JMJD6 shRNA-2 CHP134 cells after treatment with vehicle control or DOX as the new Supplementary Table 1.
- We have added the results from the new Gene Set Enrichment Analysis of genes down-regulated by JMJD6 shRNA-1 and JMJD6 shRNA-2 with p < 0.05 and adjusted p (FDR) < 0.20, as identified by the 4 replicates of microarray experiments, as the new Supplementary Table 2.
- We have revised the first paragraph on page 9 in “**Results**” section to reflect the above change.
- We have revised legends for Supplementary Tables 1 and 2 to show that the microarray experiments were repeated for four times.

Question 2. I cannot find MYCN as a differentially expressed RNA in the microarray data. Authors present experimental evidence (qRT-PCR and Western) in the revised Figure 3C that MYCN is a transcriptional target of JMJD6.

Response:

In the microarray data, *MYCN* was not among the genes down-regulated by JMJD6 shRNA-1 or JMJD6 shRNA-2. However, our RT-PCR and immunoblot analyses confirmed that N-Myc mRNA and protein were down-regulated by two independent JMJD6 shRNAs (Fig. 3c) and two independent JMJD6 siRNAs (Supplementary Fig. 3e, f). In addition, our ChIP-Seq and ChIP PCR data both revealed 60% reduction in RNA polymerase II binding at the *MYCN* gene promoter after JMJD6 knockdown (Supplementary Fig. 3b, c, d). Taken together, the data reveal that JMJD6 up-regulates JMJD6 gene transcription.

Question 3. In GSEA analysis, authors show that enrichment of NMYC-01 gene set upon JMJD6 knockdown in supplementary table 3. However, the enrichment is not significant (FDR value: 0.48).

Response:

Our microarray experiments were performed in DOX-inducible control shRNA and JMJD6 shRNA CHP134 cells after treatment with vehicle control or DOX for 40 hours. Myc or N-Myc target genes were not enriched in GSEA analysis of genes down-regulated by JMJD6 shRNAs by 1.5 fold with $p < 0.05$ and $FDR < 0.20$.

In the 1st revision manuscript, to examine whether JMJD6 directly regulates gene transcription, we performed ChIP sequencing (ChIP-Seq) experiments in DOX-inducible JMJD6 shRNA CHP134 cells, also 40 hours after treatment with control or DOX. The ChIP-Seq data identified a list of genes with considerably reduced RNA polymerase II (RNA Pol II) binding peaks at their gene promoters (Supplementary Table 4). GSEA analysis of the repressed genes showed that the top 4 gene sets were G2M checkpoint genes, E2F target genes, mitotic spindle genes and Myc target genes (Supplementary Table 5). As such, our RNA Pol II ChIP sequencing data demonstrate that JMJD6 regulates N-Myc target gene transcription.

Question 4. Overall, microarray data in the revised version is not well controlled. Information about the replicates for each microarray experiment is lacking. FDR cut off at 0.25 is too high to obtain reliable data.

Response:

As discussed in Question 1, we have repeated microarray experiments so that we now have 4 replicates of microarray data from DOX-inducible control shRNA, JMJD6 shRNA-1 and JMJD6 shRNA-2 CHP134 cells, and we have analyzed the microarray data by integrating data from DOX-inducible JMJD6 shRNA-1 and shRNA-2, filtered genes with fold changes of > 1.5 , p values < 0.05 and FDR values < 0.20 by both JMJD6 shRNA-1 and JMJD6 shRNA-2 (new Supplementary Table 1), and perform GSEA analyses (new Supplementary Table 2).

While in most papers FDR cut-off is set at 0.25, we have used FDR cut-off of 0.20 for both microarray gene expression and GSEA analysis in Supplementary Tables 1-3.

Question 5. The authors have not provided crucial data on JMJD6/MYCN (ChIP-seq) binding sites across the genome. Authors have also not addressed genome-wide E2F2 binding sites in control and JMJD6 shRNA KD cells, as reviewer 1 suggested.

Responses:

5.1 Crucial data on JMJD6/MYCN (ChIP-seq) binding sites across the genome

In the 1st revision manuscript, we have demonstrated that JMJD6 protein directly binds to N-Myc protein at the Myc Box II region. To address this question, we have performed triplicate

ChIP-Seq experiments with anti-JMJD6 and anti-N-Myc antibodies in CHP134 neuroblastoma cells. We have also performed ChIP-Seq with anti-acetylated histone H3K27 antibody to define super-enhancers and typical enhancers. Bioinformatics analysis showed that both JMJD6 and N-Myc protein bound to super-enhancers, typical enhancers as well promoters. Importantly, the majority of super-enhancer peaks (7/9) and typical enhancer peaks (65/96) bound by JMJD6 were also bound by N-Myc, and approximately 47% of promoter peaks (84/177) bound by JMJD6 were also bound by N-Myc (new Fig. 3g and new Supplementary Tables 6, 7, 8). Our data suggest that JMJD6 modulates gene transcription partly through binding to N-Myc, and that the interaction between JMJD6 and N-Myc is likely to be important for efficient gene transcription.

5.2 Genome-wide E2F2 binding sites in control and JMJD6 shRNA knockdown cells, as Reviewer 1 suggested.

We have searched the Gene Expression Omnibus website of the NCBI, and have found E2F2 ChIP sequencing dataset GSM1239497, a validation set for the ChIP sequencing SuperSeries GSE51142. The Abcam ab65222 anti-E2F2 antibody was used and the ChIP-seq experiment failed quality control (<https://www.ncbi.nlm.nih.gov/geo/query/acc.cgi?acc=GSM1239497>). The E2F2 ChIP sequencing data were therefore not included in the *Cell* paper (Yan J *et al. Cell.* 2013; 154:801-13). Unfortunately, no ChIP sequencing with any E2F2 antibody has ever been reported since then, according to the Gene Expression Omnibus website and the PubMed website.

For the 1st revision, to address Reviewer #1's question, we treated DOX-inducible JMJD6 shRNA-2 CHP134 cells with vehicle control or DOX, followed by ChIP sequencing with a control IgG or anti-E2F2 antibody (Catalogue # sc-9967, Santa Cruz Biotechnology). Unfortunately, the anti-E2F2 antibody ChIP sequencing did not work, as bioinformatics analysis revealed very few peaks in vehicle control-treated or DOX-treated samples.

For this 2nd revision, to address Reviewer #2's same question, we have treated DOX-inducible JMJD6 shRNA-2 CHP134 cells with vehicle control or DOX, followed by ChIP sequencing with a control IgG or anti-E2F2 antibody (Catalogue # DR 1095, Merck). The experiments were repeated three times. Unfortunately, the anti-E2F2 antibody ChIP sequencing did not work, as bioinformatics analysis revealed very few peaks in vehicle control-treated or DOX-treated samples. We also tried other anti-E2F2 antibodies for ChIP PCR. Unfortunately, no anti-E2F2 antibody showed efficient binding at gene promoters. In contrast, our ChIP sequencing experiments with anti-acetylated histone H3K27 antibody, anti-RNA polymerase II antibody, anti-N-Myc antibody and anti-JMJD6 antibody all worked well. We currently could not find an anti-E2F2 antibody suitable for ChIP sequencing.

Changes:

- We have added new Fig. 3g to depict the considerable overlap between peaks bound by JMJD6 and peaks bound by N-Myc in super-enhancer, typical enhancer and promoter regions, according to our new JMJD6 antibody and N-Myc antibody ChIP-Seq data in neuroblastoma cells.
- We have added new Supplementary Tables 6-8 to list signal peaks bound by JMJD6, by N-Myc, or by both JMJD6 and N-Myc in super-enhancer (Supplementary Table 8), typical enhancer (Supplementary Table 7) and promoter (Supplementary Table 6) regions, according to our new JMJD6 antibody and N-Myc antibody ChIP-Seq data in neuroblastoma cells.
- We have added the following sentences from the second last line on page 10 to the fourth line on page 11 in “**Results**” section: “To further demonstrate the JMJD6-N-Myc protein complex, we performed ChIP-Seq with anti-JMJD6 and anti-N-Myc antibodies in CHP134 cells. Bioinformatics analysis showed that both JMJD6 and N-Myc protein bound to super-enhancers, typical enhancers as well promoters. Importantly, the majority of super-enhancers and typical

enhancers bound by JMJD6 were also bound by N-Myc, and approximately 47% of promoters bound by JMJD6 were also bound by N-Myc (Fig. 3g, Supplementary Tables 6, 7, 8).....”.

- We have revised the sentences from the 2nd line to the 11th line on page 17 in “**Discussion**” section to the following: “.....Importantly, we have confirmed that JMJD6 protein forms a complex with BRD4 protein in neuroblastoma cells, that JMJD6 protein directly binds to N-Myc protein at the Myc Box II region, and that the majority of super-enhancers, typical enhancers and promoters bound by JMJD6 are also bound by N-Myc. In the literature, N-Myc regulates gene transcription mainly through binding to target gene promoters and enhancers^{14,30}, and JMJD6 activates gene transcription by interacting with MED12 in the mediator complex, interacting with BRD4 at anti-pause enhancers and binding to gene promoters^{7,31-33}. Our data therefore suggest that JMJD6 activates gene transcription partly through binding to N-Myc, and that the interaction between JMJD6 and N-Myc is likely to be important for efficient gene transcription.”

Reviewers' comments:

Reviewer #1 (Remarks to the Author):

Authors evaluated expression of JMJD6 in cases with segmental 17q gains and gains of whole chromosome 17 and concluded that elevated expression of JMJD6 is caused by segmental 17q gains but not whole chromosome 17 gains.

Critical question is how authors can conclude that among hundreds of genes on 17q (Fig. 1a), JMJD6 is a major responsible genes of 17q gain that have long been implicated in NB pathogenesis. Expression of JMJD6 in cases with 17q gains is only 2x higher than that in 17q-gain negative cases. Meanwhile, authors knockdown the JMJD6 expression more profoundly and overexpress more than 2x. No focal gains narrowly involved JMJD6, while several focal gains are found in 17q that do not include JMJD6. I believe that this paper is terribly misleading as long as it claims that JMJD6 is a major target of 17q gain in neuroblastoma. Nevertheless, this paper may be worth publication, if authors substantially attenuate their conclusion that JMJD6 is the target of 17q gain in NB. The possibility should be mentioned only briefly that copy number gain of JMJD6 might partly explain the pathogenesis of NB caused by 17q gain.

Reviewer #2 (Remarks to the Author):

The authors have done a great job with the re-revised manuscript. Overall, I am happy with the revision. I just have a one concern regarding the role of JMJD6 in the regulation of MYC and MYC target genes. Authors characterize MYCN as JMJD6 using RNAP II ChIP-seq (from JMJD6 KD cells), and RT-qPCR experiments (JMJD6 siRNA and shRNA). However, authors could not find MYCN as JMJD6 target in the microarray data from JMJD6 KD cells. The authors have clearly explained this problem in their rebuttal. It seems that microarray is not sensitive enough to pick up MYCN changes following JMJD6 KD.

Responses to Reviewers

Reviewer #1:

Authors evaluated expression of JMJD6 in cases with segmental 17q gains and gains of whole chromosome17 and concluded that elevated expression of JMJD6 is caused by segmental 17q gains but not whole chromosome 17 gains.

Critical question is how authors can conclude that among hundreds of genes on 17q (Fig. 1a), JMJD6 is a major responsible genes of 17q gain that have long been implicated in NB pathogenesis. Expression of JMJD6 in cases with 17q gains is only 2x higher than that in 17q-gain negative cases. Meanwhile, authors knockdown the JMJD6 expression more profoundly and overexpress more than 2x. No focal gains narrowly involved JMJD6, while several focal gains are found in 17q that do not include JMJD6. I believe that this paper is terribly misleading as long as it claims that JMJD6 is a major target of 17q gain in neuroblastoma. Nevertheless, this paper may be worth publication, if authors substantially attenuate their conclusion that JMJD6 is the target of 17q gain in NB. The possibility should be mentioned only briefly that copy number gain of JMJD6 might partly explain the pathogenesis of NB caused by 17q gain.

Responses:

In the previous version, we analyzed the human neuroblastoma array comparative genomic hybridization (array-CGH) datasets from 209 patients, and found that the *JMJD6* gene was gained in every human neuroblastoma tissue with 17q gain (Fig. 1a).

As suggested by Reviewer #1, we have revised the manuscript to attenuate our conclusion that JMJD6 is the target of 17q gain in neuroblastoma.

Changes:

- We have revised the **Title** of the manuscript *from* “JMJD6 gene gain is a tumorigenic factor and therapeutic target in neuroblastoma” *to* “JMJD6 is a tumorigenic factor and therapeutic target in neuroblastoma”
 - We have revised the last sentence in the **Abstract** section *from* “Our findings therefore identify JMJD6 gene gain as a neuroblastoma tumorigenesis factor, and the combination therapy as a novel treatment strategy.” *to* “Our findings therefore identify JMJD6 as a neuroblastoma tumorigenesis factor, and the combination therapy as a novel treatment strategy.”
 - We have revised the last sentence on page 4 in the **Introduction** section *from* “.....treatment with THZ1 and the histone deacetylase (HDAC) inhibitor panobinostat synergistically reduced JMJD6, E2F2, N-Myc and c-Myc expression, induced tumor cell apoptosis *in vitro* and led to neuroblastoma tumor regression in mice, which were significantly reversed by forced JMJD6 over-expression, suggesting this combination as a novel therapeutic approach for neuroblastoma characterized by 17q21-ter gain” *to* “.....treatment with THZ1 and the histone deacetylase (HDAC) inhibitor panobinostat synergistically reduced JMJD6, E2F2, N-Myc and c-Myc
-

expression, induced tumor cell apoptosis *in vitro* and led to neuroblastoma tumor regression in mice, which were significantly reversed by forced JMJD6 over-expression, suggesting this combination as a novel therapeutic approach for neuroblastoma”

- We have revised the last sentence on page 18 in the **Discussion** section *from* “Our data therefore confirm the critical role of JMJD6 in chromosome 17q21-ter-gained neuroblastoma tumorigenesis. As the CDK7 inhibitor SY-1365 is currently in clinical trials in cancer patients (<https://clinicaltrials.gov/ct2/show/NCT03134638>) and the HDAC inhibitor panobinostat is currently in clinical practice, our findings identify CDK7 inhibitor and HDAC inhibitor combination therapy as a novel treatment strategy for neuroblastoma characterized by chromosome 17q21-ter gain” *to* “Our data therefore confirm the critical role of JMJD6 in neuroblastoma tumorigenesis. As the CDK7 inhibitor SY-1365 is currently in clinical trials in cancer patients (<https://clinicaltrials.gov/ct2/show/NCT03134638>) and the HDAC inhibitor panobinostat is currently in clinical practice, our findings identify CDK7 inhibitor and HDAC inhibitor combination therapy as a novel treatment strategy for neuroblastoma.”

Reviewer #2:

The authors have done a great job with the re-revised manuscript. Overall, I am happy with the revision. I just have a one concern regarding the role of JMJD6 in the regulation of MYC and MYC target genes. Authors characterize MYCN as JMJD6 using RNAP II ChIP-seq (from JMJD6 KD cells), and RT-qPCR experiments (JMJD6 siRNA and shRNA). However, authors could not find NMYC as JMJD6 target in the microarray data from JMJD6 KD cells. The authors have clearly explained this problem in their rebuttal. It seems that microarray is not sensitive enough to pick up MYCN changes following JMJD6 KD.

Response:

We thank the Reviewer for the generous comments and agreement with our interpretation. As the reviewer is happy with the revision, no further changes have been made.